# Obligatory Role of AMPK Activation and Antioxidant Defense Pathway in the Regulatory Effects of Metformin on Cellular Protection and Prevention of Lens Opacity

**DOI:** 10.3390/cells11193021

**Published:** 2022-09-27

**Authors:** Bhavana Chhunchha, Eri Kubo, Dhirendra P. Singh

**Affiliations:** 1Department of Ophthalmology and Visual Sciences, University of Nebraska Medical Center, Omaha, NE 68198, USA; 2Department of Ophthalmology, Kanazawa Medical University, Ishikawa 9200293, Japan

**Keywords:** antioxidants, aging, oxidative stress, Peroxiredoxin 6, Metformin, age-related diseases, Bmal1, Nrf2, antioxidant response

## Abstract

Increasing levels of oxidative-stress due to deterioration of the Nrf2 (NFE2-related factor)/ARE (antioxidant response element) pathway is found to be a primary cause of aging pathobiology. Metformin having anti-aging effects can delay/halt aging-related diseases. Herein, using lens epithelial cell lines (LECs) of human (h) or mouse (m) and aging h/m primary LECs along with lenses as model systems, we demonstrated that Metformin could correct deteriorated Bmal1/Nrf2/ARE pathway by reviving AMPK-activation, and transcriptional activities of Bmal1/Nrf2, resulting in increased antioxidants enzymatic activity and expression of Phase II enzymes. This ensued reactive oxygen species (ROS) mitigation with cytoprotection and prevention of lens opacity in response to aging/oxidative stress. It was intriguing to observe that Metformin internalized lens/LECs and upregulated OCTs (Organic Cation Transporters). Mechanistically, we found that Metformin evoked AMPK activation-dependent increase of Bmal1, Nrf2, and antioxidants transcription by promoting direct E-Box and ARE binding of Bmal1 and Nrf2 to the promoters. Loss-of-function and disruption of E-Box/ARE identified that Metformin acted by increasing Bmal1/Nrf2-mediated antioxidant expression. Data showed that AMPK-activation was a requisite for Bmal1/Nrf2-antioxidants-mediated defense, as pharmacologically inactivating AMPK impeded the Metformin’s effect. Collectively, the results for the first-time shed light on the hitherto incompletely uncovered crosstalk between the AMPK and Bmal1/Nrf2/antioxidants mediated by Metformin for blunting oxidative/aging-linked pathobiology.

## 1. Introduction

A progressive loss of antioxidant response and thereby acceleration of oxidative stress, resulting in deranged cellular physiology during aging are the primary risk factors for the etiology of a variety of pathologies related to the development of aging diseases, including blinding diseases [1,2,3]. It is now consensus in the field of redox biology that increasing oxidative stress during aging is a primary culprit in the etiopathobiology of aging disorders. With advancing age, oxidative stress is amplified due to the generation of oxidant molecules from external sources as well as several intracellular sources lacking antioxidants, the primary line of defense. However, several studies now have identified that Nrf2/ARE antioxidant pathway is a major player in maintaining the cellular redox homeostasis against oxidative stress and other deleterious insults as well as a decline of Nrf2 signaling leading to aging diseases [4,5,6,7]. Nrf2/ARE pathway comprises several antioxidant genes, such as catalase (Cat), glutathione-peroxidase (GPxs), hemeoxygenase 1 (HO1) and peroxiredoxin 6 (Prdx6), NAD(P)H quinone oxidoreductase (NQO1), including other PhaseI or PhaseII enzymes. Notably, the promoters of these antioxidants contain the Nrf2 regulatory site, ARE [8]. At normal physiological conditions (reducing environment), basal activity of Nrf2 is controlled by cytoplasmic Kelch-like-ECH-associated protein 1 (Keap1), an adaptor protein for CULL3-based ubiquitin E3 ligase that continuously ubiquitinates Nrf2 for proteasomal degradation via 26S proteosome [9,10]. Upon oxidative stimuli or electrophiles exposure, Nrf2-Keap1 interaction is dislodged in a dose-dependent fashion, and in turn leads to Nrf2 translocation to the nucleus, where it transactivates transcription of genes by binding to AREs [5]. However, Nrf2 activity can be regulated at transcriptional and translational levels as well as interactions with other proteins [11,12]. Recent evidence discloses that not only Nrf2 deficiency but a deficiency of the core clock transcriptional protein Bmal1 is linked to the etiology of aging-related degenerative diseases [13,14,15]. It has been found that the regulation of genes, including antioxidant genes other than clock genes, can be regulated by Bmal1 [16,17,18].

Importantly, there is now a growing body of extensive research in the redox biology field showing evidence that several genes, including Nrf2 and major Nrf2 antioxidants, can directly be activated by core clock transcriptional protein Bmal1 by binding to its response elements, E-Box elements, present in the regulatory region of their promoter. This demonstrates that Bmal1 plays a critical role to stabilize cellular redox homeostasis [17,18]. Studies in humans and mice have shown that deficiency of Bmal1 leads to aging pathologies, leading to disease states, like age-related cataracts (ARC) [14,15,19], while its increased abundance extends the life span and health span [20]. Very recently, our group has shown that the cooperativity of Bmal1 and Nrf2 is required for increased transcription of antioxidant genes and to maintain redox homeostasis of aging lens cells and their protection against oxidants [21]. We have reported that the most of major antioxidant genes and Nrf2 have functional Nrf2/ARE as well as Bmal1/E-Box, and the binding and transactivation potential of these transcriptional proteins depend on their cellular abundance in response to oxidative stress within the cellular microenvironment [21]. Additionally, we have also observed that Bmal1, a regulator of Nrf2 and Nrf2 antioxidants, declines with aging, resulting in the loss of Nrf2 and Nrf2 antioxidants, and those cells were more vulnerable to death. This suggests that Bmal1 is a critical element to maintain Nrf2 and Nrf2 antioxidant pathways. With loss-and gain-of-activity experimentations, we found the evidence that Bmal1-Nrf2-antioxidant axis is crucial to defend aging or oxidative related injurious signaling. Previously, it has been shown that Bmal1 controls IL-1β via Nrf2 regulation and thereby, maintains cellular homeostasis by regulating reactive oxygen species (ROS) levels [13,18,22,23]. Fascinatingly, several recent studies have shown that AMP-activated kinase signaling is linked to a decline in the capacity to respond to redox state or metabolic changes with advancing age [8,24,25,26]. Notably, it also has been illustrated that AMPK activation can regulate Bmal1 and Nrf2 expression and activity [27,28,29,30]. Additionally, it has been observed that AMPK activation derepressed Bmal1 activity by destabilizing CRY protein [27,31].

AMP-activated protein kinase (AMPK), a major sensor of cellular energy status, is highly conserved across the species and is activated in response to metabolic stress like oxidative stress, inflammation, and multiple upstream stimuli [32,33]. AMPK is present as a trimeric form having a catalytic subunit, α subunit, and two regulatory subunits, β and γ subunits. The expression levels and patterns of AMPK can vary across tissues, resulting in different subunit combinations in different cell types with redundant functions [34]. Stress-inducible, AMPK is a master regulator of cellular physiology under normal physiological conditions as well as during internal and external stresses. Recent advancements in research show that activated AMPK is involved in the epigenetic remodeling of promoters and regulates gene transcription [35]. In addition, AMPK has been reported to directly regulate Nrf2 by phosphorylating at residues, serine 550, which results in nuclear accumulation of Nrf2 and subsequent increased expression of antioxidants gene with further increased in Nrf2 expression (feed-forward process via auto regulation of Nrf2) [5,36,37,38,39,40,41]. Recently, it also has been documented that AMPK activation amplifies transcription of Nrf2 antioxidant gene by suppressing BTB Domain and CNC homolog 1 (Bach1), a competitor of Nrf2 at ARE sequences. It is note-worthy that a great deal of evidence indicates a role for the AMPK in the regulating expression and activity of core clock transcription factors, such as Bmal1 [18,21,27,28,32,42]. Thus, AMPK activation-dependent Bmal1 and Nrf2 expression and their cooperative regulation of antioxidant genes in favor of cellular defense suggest that AMPK activation via AMPK activator(s), should be an effective strategy to prevent/delay aging or oxidative stress-related pathobiology. However, given the therapeutically beneficial effects associated with activation of AMPK, many AMPK activators, such as Sulforaphane, Metformin, resveratrol, berberine as well as AMP mimetics (AICAR) or AMPK binding compounds, etc. have been well recognized and examined through in vitro and in vivo experimentation [32,43,44,45,46,47,48,49]. However, many drugs have been identified as AMPK activators, we selected repurposing FDA-approved Metformin for the present study as it met all the above-stated criteria (AMPK/Bmal1/Nrf2/antioxidant axis) to test its efficacy in reactivation of Bmal1/Nrf2/antioxidant defense pathway and its potential in protecting LECs/aging LECs or organ (eye lenses) against oxidative stresses. We discovered that Metformin is safe and FDA approved has anti-aging properties, and has the ability to block aging-related pathobiology where it enhances health-span and lifespan. These properties of Metformin are extensively illustrated and reviewed [41,50,51,52,53,54,55].

Aging-related pathologies share a series of the common denominator(s) or pathogenic signaling and studies suggest that delaying/treating one age-related pathobiology can stand with others [56,57,58,59,60,61]. Hence, using eye lens, and cell lines of LECs along with primary LECs-derived from humans and mice, one of the elegant model systems for exploring the molecular mechanisms of age-related adverse signaling and its plausible prevention by therapeutic molecule [62,63], herein we uncovered that oxidative stress/aging-related deterioration of Bmal1/Nrf2/antioxidant genes pathway can be reactivated by Metformin. Data revealed that Metformin is internalized in the lenses/LECs through organic transporter (OCTs). Unexpectedly, we observed that Metformin augmented the expression of all three OCTs levels in LECs, suggesting that it promotes its accessibility into cells/tissues. Using loss-and gain-of-function studies and transactivation experiments, we identified that Metformin increased DNA binding activities of Bmal1 to E-Box and Nrf2 to ARE in the antioxidant gene promoters, like Prdx6, and both sites were essential elements for the optimum expression of antioxidant genes. Additionally, we illustrated that Metformin defended aging LECs and delayed/prevented lens opacity against exogeneous oxidant(s) by reactivating the AMPK/Bmal1/Nrf2/antioxidants axis. We suggest that Metformin provides protective mechanisms to shield lens/LECs, otherwise, prone to an adverse redox environment that elicits aging-related pathologies, in turn leading to disease states, like ARC. Thus, our study provides proof of concept that Metformin can be seen as a future therapeutic candidate to treat/prevent ARC, including age-related pathobiology in general.

## 2. Materials and Methods

### 2.1. Human Lens Epithelial Cells (hLECs), SRA-hLECs, Primary Emb-hLECs and hLECs Culture and Maintenance

Three types of human Lens epithelial cell (hLECs) were used: (1) hLECs cell line (SRA01/04) immortalized with SV40 [64], (2) primary human embryonic LECs and (3) primary human LECs isolated from deceased persons of variable ages. To avoid confusion, the remaining text will designate the immortalized LECs as SRA-hLECs, and embryonic human lens epithelial cells as Emb-hLECs and the primary human LECs of different ages as hLECs or primary hLECs.

The SRA-hLECs (a kind gift of late Dr. Venkat N. Reddy, Oakland University, Rochester, MI, USA) [64] were maintained routinely in our laboratory. Briefly, cells were cultured in a 75-mm tissue culture flask in Dulbecco’s Modified Eagle Medium (DMEM, Invitrogen, Waltham, MA, USA) supplemented with 15% heat-inactivated fetal bovine serum (FBS, Atlanta Biologicals, Atlanta, GA, USA), 100 µg/mL streptomycin, and 100 µg/mL penicillin in a water-jacketed incubator maintained at 37 °C with 5% CO_2_ as described previously [65,66].

Primary embryonic human lens epithelial cells (Catalog # 6550) were purchased from ScienCell Research Laboratories (Carlsbad, CA, USA) and designated as Emb-hLECs. These cells were maintained following the company’s protocol.

Primary hLECs were isolated from normal eye lenses of deceased persons or healthy donors of different ages (17 y,19 y, 19 y, 20 y, 23 y, 23 y, 23 y, 24 y, 25 y, 53 y, 53 y, 54 y, 54 y, 55 y, 55 y, 58 y, 59 y, 65 y, 66 y, 67 y, 71 y, 71 y, 73 y, 74 y, 74 y, 76 y, 76 y, 78 years (y) of old) procured from the Lions Eye Bank, Nebraska Medical Center, Omaha, NE, and National Development & Research Institute (NDRI), Inc., Philadelphia, PA, USA. LECs isolated from these lenses were used for the present study. However, according to regulation HHS45CFR 46.102(f), studies involving lenses or LECs from deceased subjects are not considered human subject(s) research as stated under 45CFR46.102(f) 10(2) and, therefore, do not require IRB oversight. However, due to the limited sample size, eye lenses were grouped by age: lenses age and number; 17 to 25 y, n = 8; age 53 to 59 y, n = 10; and age 65 to 78 y, n = 10. To isolate LECs from the lenses, the capsule was cut, and the capsule containing LECs was explanted in 35 mm culture dishes coated with collagen IV containing DMEM medium supplemented with 15–20% FBS as described previously [67,68,69,70]. Culture explants were trypsinized and isolated cells were cultured in Petri dishes containing a complete medium. Cell cultures showing 90 to 100 percent confluency were harvested and used for the study [66,71,72]. A specific marker, αA-crystallin, was used for validation of LECs identity (data not shown).

### 2.2. C57BL/6 Mouse and Mouse Lens Epithelial Cells (mLECs) Isolation from Lenses

C57BL/6 mice of variable ages were obtained from Charles River laboratories, Wilmington, MA, USA, and were maintained at a stable temperature (22 ± 2 °C) and humidity (55 ± 5%). The mice were maintained under specific pathogen-free conditions in an animal facility. For the experiments, the mice were sacrificed, and lenses were isolated. The lens capsule containing LECs was gently taken out and then explanted in culture dishes precoated with collagen IV containing complete DMEM (10 to 15% FBS) [67,68,69,70]. Cell cultures showing 90 to 100% confluency were harvested and used for the experiments [66,71,72,73]. Following, mLECs were used in this study: (1) a cell line of mLECs (2) primary mLECs isolated from C57BL/6 mice of both sexes.

All animal studies followed the recommendations set forth in the “Statement for the Use of Animals in Ophthalmic and Visual Research” by the Association for Research in Vision and Ophthalmology and were approved by the Institutional Animal Care and Use Committee (IACUC), University of Nebraska Medical Center (UNMC), Omaha, NE, USA.

### 2.3. Quantitation of Intracellular Reactive Oxygen Species (ROS) by H2-DCF-DA in Lenses and LECs

Intracellular redox state levels of lenses isolated from C57BL/6 mice of different ages (6 months (M), 16 M and 22 M), were immediately frozen at −80 °C to estimate ROS levels according to our published protocol [21,74]. In brief, lenses were homogenized (100 mg/mL) in freshly prepared homogenization buffer (50 mM Phosphate buffer containing 0.5 mM PMSF, 1 mM EDTA, 1 µM Pepstatin, 80 mg/L Trypsin Inhibitor, Ph 7.4). The same amount of homogenate was added to 96-well cell culture plate. To quantify the levels of ROS, H2-DCF-DA dye was added to the wells to achieve a final concentration of 30 µM. After 30 min, intracellular fluorescence was detected at O.D., Ex485 nm/Em530 with a Spectra Max Gemini EM (Molecular Devices, San Jose, CA, USA). Furthermore, the intracellular levels of ROS in LECs were quantified by H2-DCF-DA, as noted above and following our previously published method [2,75,76,77].

### 2.4. RNA isolation and mRNA Analysis of Different Ages of Mouse Lenses and Mouse or Human LECs Using RT-qPCR

To monitor the status of antioxidant pathway-related gene expression during aging and the effects of Metformin on their regulation, total RNA was isolated from lenses/LECs -derived from C57BL/6 mice of both sexes (6 M, 16 M and 22 M) as well as human aging LECs isolated from lenses of subjects of different age groups, by using the single-step guanidine thiocyanate/phenol/chloroform extraction method (Trizol Reagent, Invitrogen, Waltham, MA, USA) according to the manufacturer’s specifications and as described previously [5,21,74]. Total RNA (0.5 to 5 micrograms) was converted to cDNA with Superscript II RNAase H-reverse-transcriptase to assess the expression levels of Bmal1, Nrf2, Prdx6, GPx1, SOD1, GST*π*, Catalase, HO1, NQO1, GCLC and GCLM, OCT1, OCT2, OCT3 using the primers specific to corresponding genes. Real-time quantitative PCR (RT-qPCR) was carried out with SYBR Green Master Mix (Roche Diagnostic Corporation, Indianapolis, IN, USA) in a Roche^®^ LC480. Sequence detector system (Roche Diagnostic Corporation) under PCR conditions of 5–10 min hot start at 95 °C, followed by 40–45 cycles for 10 s (sec) at 95 °C, 30 s at 60 °C and 10 s at 72 °C. Sequences of the primers used in the study are shown below in Table 1.

The relative quantity of the mRNA was obtained using the comparative threshold cycle (CT) method. The expression levels of target genes were normalized to the levels of β-actin as an endogenous control in each group.

### 2.5. Assay for Phospholipase A_2_ (PLA_2_) Activity

Phospholipase A_2_ (PLA_2_) activity was determined according to the manufacturer’s protocol (EnzChek Phospholipase A2 kit; E10217, Invitrogen, Waltham, MA, USA) and our published protocol [75,78]. In brief, total protein isolated from mouse/human lenses/LECs of different ages or LECs untreated or treated with Metformin as indicated in figures and legends was quantified by Bicinchoninic acid (BCA) protein assay (Thermo Fisher Scientific, Waltham, MA, USA). To prepare the standard curve, different concentrations (0–10 Units/mL) of PLA_2_ were prepared by diluting PLA_2_ stock solution (500 Units/mL) with 1× reaction buffer up to 50 µL. An equal amount of protein isolated from mouse/human lenses/LECs and LECs was diluted with 1× PLA_2_ reaction buffer to make volume up to 50 µL, then 50 µL of the substrate-liposome mix were added to each microplate well-containing control, standard and the samples to start the reaction with 100 µL total volume. The fluorescence units were measured at optical density (O.D.), Ex485nm/Em535 nm using a microplate reader (DTX 880, Multimode Detector, Molecular device, San Jose, CA, USA) as shown in the figures.

### 2.6. Assay for Glutathione Peroxidase Activity

Glutathione (GSH) peroxidase activity in Metformin-treated or untreated mLECs or hLECs was estimated according to manufactures protocol (Glutathione Peroxidase activity kit, Cat No. ADI-900-158, Enzo Life Sciences, Farmingdale, NY, USA) and our published protocol [75,78]. Total cell lysate was prepared, and protein was quantified by the BCA protein assay method. An equal amount of protein was used to measure the enzymatic activity. To set up the reaction, 140 µL of 1× assay buffer and 20 µL of 10× reaction mix and 20 µL of standard (Glutathione Peroxidase) or samples or control were added to 96-well plates. Then, the reaction was initiated by quickly adding 20 µL of cumene hydroperoxide to each well and O.D. was measured at absorbance 340 nm every 1 min up to 15 min period. Blank O.D was subtracted from the standard as well as sample O.D. to obtain the net rate of absorbance at 340 nm for the GSH peroxidase activity.

### 2.7. Assay for Catalase Activity

Catalase activity was measured following the manufacturer’s manual (Catalase Assay, Kit, LS-K245, LifeSpan BioSciences, Inc., Seattle, WA, USA). For samples, 10 µL of an equal amount of total protein isolated from mouse/human lenses/LECs of different ages or LECs untreated or treated with Metformin were separately added into wells of 96-well plates. To initiate the Catalase reaction 90 µL of freshly prepared 50 µM H_2_O_2_ substrate (1 µL of the 4.8 mM H_2_O_2_ with 95 µL assay buffer) was added to the blank, control and sample wells. 96-well plate was incubated at room temperature for 30 min. After a quick mix, the plates were submitted for detection. For Catalase standard curve, different concentrations of H_2_O_2_ (0, 120 µM, 240 µM and 400 µM) were prepared, and then 10 µL of each standard and 90 µL of assay buffer were added into separate wells of 96-well plates and proceeded for detection. For detection, 100 µL of detection reagent (mixture of 102 µL assay buffer, 1 µL Dye reagent, 1 µL HRP enzymes) was added to each reaction well of standard, samples and controls and incubated for 10 min. After mixing, optical density (O.D.) was measured at 570 nm with DTX 880 Multimode Detector (SpectraMAX Gemini, San Jose, CA, USA) and Catalase activity was presented as U/L.

### 2.8. Assay for SOD Activity

The levels of SOD activity were determined according to the manufacturer’s manual instructions (Superoxide Dismutase (SOD) Assay, Kit, LS-K224, LifeSpan BioSciences, Inc., Seattle, WA, USA). For the standard curve, the SOD enzyme was diluted using diluent and different concentrations (0–3 U/mL) of SOD. Standards were prepared and 20 µL was added to a 96-well microplate. For samples, total protein isolated from Metformin-treated or untreated human or mouse aging lenses**/**LECs was quantified using the BCA protein assay method. 20 µL of samples containing equal amounts of protein and dilutions were added to separate wells of a 96-well plate. 160 µL of working reagent (mixture of 160 µL assay buffer, 5 µL Xanthine and 5 µL WST-1) was added to the standard, samples, and control wells. After mixing the contents by tapping the plate, 20 µL of diluted XO enzyme (1:20 diluted in diluent) was added and immediately the O.D. was measured at 440 nm and 0 min (OD_0_) with DTX 880 Multimode Detector (SpectraMAX Gemini, San Jose, CA, USA). The plate was incubated for 60 min at room temperature in dark and again the O.D. was measured at 440 nm (OD_60_). Each standard and sample wells were calculated using the following formula: ΔOD_60_ = OD_60_ − OD_0_ and presented in percentage.

### 2.9. Cell Viability Assay (MTS Assay)

Cell viability of Metformin (Catalog no. 317240, Calbiochem, Millipore Sigma, Burlington, MA, USA) treated or untreated LECs facing H_2_O_2_-induced oxidative stress was evaluated using a colorimetric MTS assay (Promega, Madison, WI, USA) according to the manufacturer’s method and as described previously [3,76]. This assay of cell proliferation/viability uses 3-(4,5-dimethylthiazol-2-yl)-5-(3-carboxymethoxyphenyl)-2 to 4-sulphophenyl) 2H-tetrazolium salt. Upon addition to the medium containing viable cells, MTS is reduced and converted to a water-soluble formazan salt. The O.D., 490 nm value was measured after 2 h of incubation with a plate reader (DTX 880, Multimode detector, SpectraMAX Gemini, CA, USA). The data obtained were normalized with the absorbance of the untreated control(s).

### 2.10. Protein Isolation and Western Blot Analysis

Metformin-treated or untreated mouse or human LECs in the presence or absence of an AMPK inhibitor, compound C (CC; Catalog no. 171260, EMD Millipore Corporation, Burlington, MA, USA) for 24 h were subjected to protein extraction. Total cell lysates of LECs were prepared in ice-cold radioimmune precipitation buffer (RIPA buffer) and protein blot analysis was performed as described previously [5,19,22,23,24]. The membranes were probed with primary antibodies specific to the molecules to be examined: anti-Bmal1 (sc-365645, Santa Cruz Biotechnology, Dallas, TX, USA), Anti-Nrf2 (ab62352, Abcam, Cambridge, MA, USA), Anti-Prdx6 antibody (Ab Frontier, Seoul, South Korea), AMPKα (#2532S, Cell Signaling Technologies, Danvers, MA, USA), phospho AMPKα (#4188S, Cell Signaling Technologies, Danvers, MA, USA), internal controls; Tubulin (ab44928, Abcam, Cambridge, MA, USA) and LaminB1 (ab133741, Abcam, Cambridge, MA, USA) to monitor corresponding molecules expressions. Following incubation with the primary antibody, a secondary antibody (sc-2357 and sc-516102, Santa Cruz Biotechnology, Dallas, TX, USA) was added, and protein bands on the membrane were visualized by incubating the membrane with luminol reagent (sc-2048; Santa Cruz Biotechnology, Dallas, TX, USA) and images were recorded with a FUJIFILM-LAS-4000 luminescent image analyzer (FUJIFILM Medical Systems Inc., Hanover Park, IL, USA).

### 2.11. Luciferase Reporter Assay with Promoter Containing 3× Antioxidant Response Element (ARE)

SRA-hLECs or different ages of primary hLECs were transfected with pRBGP2 (3× ARE-LUC) containing three ARE sites or its mutant pRBGP4 (3× mut ARE- LUC) plasmids, a kind gift from Hozumi Motohashi, Japan) [79] along with *Renilla*, pRL-TK vector (Promega, Madison, WI, USA) using a Neon transfection system (Invitrogen, Waltham, MA, USA). After 12–14 h post-transfections, cells were washed and treated with Metformin for 24 h, luciferase activity was measured using Dual-Glo luciferase assay system with a 96-well plate (Promega, Madison, WI, USA) submitted to a microplate reader (DTX 880, Multimode Detector, Molecular device, San Jose, CA, USA).

### 2.12. Extraction of Nuclear and Cytosolic Fractions

Nuclear or cytosolic fraction was isolated following previously published methods [2,78,80]. Briefly, SRA-hLECs (1 × 10^6^) were cultured in 100-mm plates and 24 h later these cells were treated or untreated with Metformin. After washing with chilled phosphate-buffered saline (pH 7.4) the cells were collected by centrifugation. The pellet obtained was suspended in 5 pellet volumes of cytoplasmic extraction buffer [(10 mM HEPES (adjusted pH at 7.9), 0.1 mM EDTA, 10 mM KCl, 0.4% (*v*/*v*) Nonidet P-40, 1 mM DTT, 0.5 mM phenylmethylsulfonyl fluoride (PMSF) and Protease inhibitor]. After a brief incubation on ice and centrifuged at 10,000 rpm for 10 min, the cytoplasmic extract was removed from the pellet and transferred to a fresh tube. Following washing with cytoplasmic buffer without detergent, the fragile nuclei were resuspended in nuclear extract buffer (20 mM HEPES (adjusted pH at 7.9), 1 mM EDTA, 0.4 M NaCl, 10% (*v*/*v*) glycerol, 1 mM DTT, 0.5 mM PMSF and Protease Inhibitor) and subjected for 2 h at 4 °C with continuous vortexing. Finally, the extract was spun at 14,000 rpm for 15 min to pellet the nuclei. After centrifugation, the nuclear extract was aliquoted, and aliquots were stored at −80 °C. Protein was estimated according to the Bradford protein assay method and the extract was used for experiments.

### 2.13. Nrf2 Transactivation Assay

To assess the Nrf2 status and potential of Metformin activation of Nrf2 in LECs, we used a commercially available kit (TransAM Nrf2 Transcription Factor Assay Kit, Cat. No. 50296, Active motif, Carlsbad, CA, USA) and methods as described earlier [21,74,81]. Briefly, 10 µg of nuclear extract prepared from Metformin-treated or untreated SRA-hLECs was added to the strips well, followed by the addition of 40 µL complete binding buffer containing 20 pmol of the oligos containing ARE (antioxidant response element) or its mutant, mutated at the ARE site, to each sample well. 10 µL of complete lysis buffer was used as blank (control). The strips containing the samples and control wells were incubated at room temperature (RT). After 1 h, the wells were washed and the 100 µL primary antibody (1:1000 in binding buffer) was added and incubated at RT for 1 h, and then 100 µL anti-rabbit HRP conjugated antibody (1:1000 dilution) was added after three washing and further incubated for 1 h at RT. The reaction was developed by the addition of 100 µL of developing a solution to wells. The reaction was stopped by adding 100 µL of stop solution, and a reading was taken at O.D. 450 nm.

### 2.14. In-Vivo DNA Binding Assay: Chromatin Immunoprecipitation (ChIP) Experimentation

ChIP analysis was conducted by using the ChIP-IT^®^ Express kit (Cat. No. 53008; Active Motif, Carlsbad, CA, USA) and ChIP-IT^®^ qPCR analysis kit (Cat. No. 53029; Active Motif, Carlsbad, CA, USA) according to company’s protocol and as described previously [5,77,81]. Briefly, cells were processed following the protocols and the fixation reactions were halted by the addition of Glycine Fix-Stop solution. Finally, cells were collected, and the cell pellet was disrupted in 1 mL ice-cold lysis buffer using a Dounce homogenizer. After centrifugation, released nuclei were resuspended in shearing buffer. Chromatin was then disrupted to get 200–300 bp using an ultrasonic cell disruptor (Microson, Farmingdale, NY, USA). Chromatin samples were used to assess DNA and transcription factor(s) enrichment by using control IgG, an antibody specific to Nrf2 (ab62352, Abcam, Cambridge, MA, USA) and an antibody specific to Bmal1 (sc-365645, Santa Cruz Biotechnology, Dallas, TX, USA) to pull down the specific DNA fragment and then followed by RT-qPCR reaction. RT-qPCR was performed at 2 min at 95 °C, 15 s at 95 °C, 20 s at 58 °C and 20 s at 72 °C for 40 cycles in 20 μL reaction volume (RT-qPCR). The results derived from RT-qPCR are presented as histograms. Furthermore, RT-PCR was also conducted with 5 μL of DNA sample(s) using primers; Sense primer: 5′-CAGAGTCAAACCTGGCGCATC-3′ and antisense primer: 5′-CATCCTTCAGACACTATAGGCC-3′. The program used for amplification was 3 min at 94 °C, 20 s at 95 °C, 30 s at 59 °C and 30 s at 72 °C for 36 cycles. The product obtained with RT-PCR was run on 1% agarose gel, and images of transcripts were imaged using FUJIFILM-LS-4000 (FUJIFILM Corporation, Tokyo, Japan).

### 2.15. Preparation of Prdx6 Promoter-Chloramphenicol Acetyltransferase (CAT) Reporter Vector

The genomic human DNA was used to construct pCAT-hPrdx6 promoter plasmid as reported previously in our published protocol [21,81]. In brief, the genomic DNA was subjected to genomic-PCR with region-specific primers, and the 5′-flanking regions spanning from −918 to +30 bp were isolated by using an Advantage^®^ Genomic PCR Kit (Cat. No. 639103 & 639104, Clontech Laboratories, Inc., Mountain View, CA, USA). PCR product obtained was amplified and verified by sequencing as described previously [5,66]. To prepare pCAT-linked hPrdx6 promoter plasmid, the DNA fragment (−918 to +30 bp) was ligated into the basic pCAT vector (Promega) at the *SacI* and *XhoI* sites. Primers used were as follows: Forward primer; 5′-GACAGAGTTGAGCTCCACACAG-3′; and Reverse primer; 5′-CACGTCCTCGAGAAGCAGAC-3′.

### 2.16. Site-Directed Mutagenesis (SDM)

To generate site-directed specific mutation of binding sites of Bmal1/E-Box (5′−341/−336-3′; T changed to A and A changed to T) and Nrf2/ARE (5′−357/−349-3′; TG changed to GT) present in Prdx6 gene promoter (Prdx6-CAT plasmid), we utilized PCR-based site-directed mutagenesis (Catalog No. 210518; QuikChange^TM^ lightning Site-Directed Mutagenesis kit, Agilent Technologies; Santa Clara, CA, USA) following the company’s protocol and our published method [5,21,81]. The double-stranded *Prdx6* promoter construct (−918/+30) was used as template DNA with a pair of complementary primers to mutate the *Prdx6* promoter construct at E-Box and ARE sites. The primers used for mutation were as follows:

#### 2.16.1. Nrf2/ARE SDM Primer

The changed nucleotides are in red boldface type and underlined:

Forward primer: 5′-CCAGGGGGCAACG**GT**ACCGAGCCCCGCATCACGTGTGC-3′

Reverse primer: 5′-GCACACGTGATGCGGGGCTCGGT**AC**CGTTGCCCCCTGG-3′

#### 2.16.2. Bmal1/E-Box SDM Primer

Forward primer: 5′-GAGCCCCGCATC**T**CG**A**GTGCAGAGACGGC-3′

Reverse primer: 5′-GCCGTCTCTGCAC**T**CG**A**GATGCGGGGCTC-3′

### 2.17. shRNA Nrf2 Knock Down Experiment

Control *sh*RNA (sc-108060) and Nrf2 *sh*RNA (sc-37030-SH) plasmids were purchased from Santa Cruz Biotechnology, Dallas, TX, USA. SRA-hLECs were transfected with Control s*h*RNA and Nrf2 *sh*RNA. To make stable cell lines transfectants were selected with the selection marker puromycin. SRA-hLECs were transfected by using the Neon Transfection System (Invitrogen, Waltham, MA, USA) as described in our published protocol [21,74,81].

### 2.18. Bmal1 Knock down Experiment

SRA-LECs were infected with copGFP control lentiviral particle (LV *sh*-Control, sc-108084) and Bmal1 (Bmal1)/GFP *sh*RNA (LV *sh*-Bmal1, sc-38165-VS) as described in the company’s protocol (Santa Cruz Biotechnology, Dallas, TX, USA) as well as following our previously published method [21]. In brief, cells were cultured in a 12-well plate containing a complete medium. 24 h later, the complete medium was removed and a 1 mL medium containing polybrene (sc-134220, Santa Cruz Biotechnology, Dallas, TX, USA) at a final concentration of 5 µg/mL was added to culture wells. These culture-wells received the *sh*-Control or *sh*-Bmal1 lentiviral particles and mixed by swirling and then the plate was incubated overnight. After 24 h polybrene-containing medium was replaced with a complete medium (without polybrene). The infectants were treated with puromycin dihydrochloride (sc-108071, Santa Cruz Biotechnology, Dallas, TX, USA) selection marker to get stably infected cell line(s). These stably infected SRA-hLECs were utilized to conduct the experiments.

### 2.19. Isolation and Quantification of Metformin in Lens Using LC-MS/MS

LC-MS/MS analysis was carried out with the kind help of Pankaj Singh (Eppley Institute, University of Nebraska Medical Center, Buffet Cancer Center, UNMC, Omaha, NE, USA). Briefly, to analyze the level of Metformin, treated or untreated lenses were homogenized into 4 volumes of 90% methanol. Then, lens homogenate (100 µL) was mixed with methanol (90 µL) containing internal standard (IS), and 10 µL of 50% methanol in H_2_O without IS. Samples were vortexed and centrifuged at 17,000× *g* for 10 min, and then 100 µL supernatant (sample) was mixed with 50 µL H_2_O before LC-MS/MS analysis. Different concentrations of Metformin (0.5–1000 ng/mL) were used as a standard. Finally, to assess Metformin levels in the lens, 10 µL of the prepared sample(s) as noted above was loaded onto an HlLlC column fitted with an LC-MS/MS Waters Acquity UPLC system coupled to an Applied Biosystems 4000 Q TRAP quadrupole linear ion trap hybrid mass spectrometer (Applied Biosystems, MDS Sciex) as reported previously [82,83,84]. Mobile phase A consisted of 7.5 mM ammonium bicarbonate (pH 7), while mobile phase B had methanol, and the adjusted flow rate was 0.25 mL/mins. For the separation of Metformin compound, the initial mobile phase composition was 50% B for the first 4.25 min, increased to 95% B over 0.25 min, and then it was constant for 1 min. Furthermore, mobile phase B was then further reset to 50% over 0.25 min, and the column was equilibrated for 1 min before the next loading. The mass spectrometer was run at the positive ion mode using multiple reaction monitoring. The transitions were monitored; *m*/*z* 130→71.1 and recorded, and the result was calculated and presented.

### 2.20. Lens Organ Culture, Metformin and H_2_O_2_ Treatment

Lenses isolated from 14 month old mice were cultured in a 48-well plate, and then treated or untreated with 3 mM Metformin. After 24 h, these lenses were subjected to H_2_O_2_-induced oxidative stress. The lenses were observed routinely. 44 h and 96 h later lenses were photographed using Nikon SMZ 745T fitted with Nikon Camera and Computer loaded with analyses software program [67,68,78,85]. After 44 h, lenses were used to measure ROS level and RNA isolation.

### 2.21. Statistical Analysis

All experiments were independently repeated a minimum of three times. Results are presented as the mean ± standard deviation (S.D.). Statistical analysis was carried out using Student’s t-test and**/**or one-way ANOVA. Statistical significance between the control and treatment group was assessed and *p* value < 0.05 and< 0.001 was considered statistically significant.

## 3. Results

### 3.1. Aging Mouse Lenses Showed Increased ROS Accumulation, which Was Directly Associated with Reduction in Bmal1, Nrf2 and Nrf2 Antioxidant Genes Expression

Previously, we have shown that loss of Bmal1/Nrf2/antioxidant genes pathway in aging lenses/LECs is linked to higher levels of oxidative load and cellular damage [5,21,74,81]. We also have shown that cause of ROS amplification and deranged cellular physiology was specifically related to a significant reduction in Bmal1 and Nrf2 expression and activity in hLECs isolated from lenses of subjects of variable ages [5,21]. In this study, using different ages of mouse lenses, we sought to elucidate a connection between ROS levels and the Bmal1, Nrf2 and phase II antioxidants expression and activity in aging lenses, if any, as shown in Figure 1. Toward this, we quantified the total intracellular ROS using H2-DCF-DA dye and measured the expression of Bmal1, Nrf2, and Bmal1-E-Box and Nrf2-ARE regulation-dependent Phase II antioxidant genes, such as Prdx6, Gpx1, SOD1, GSTπ, Catalase, HO1, NQO1, GCLC and GCLM with RT-qPCR using their specific probes as noted in Materials and Methods section. Data revealed a progressive and significant reduction in Bmal1, Nrf2 and antioxidant genes with the dramatic accumulation of ROS in lenses, in an age-dependent manner as shown in Figure 1. Similarly, we found reduced protein expression of Bmal1, Nrf2 and their target genes in mouse aging lenses (data not shown) as reported previously [2,5]. Results clearly indicated that the dysregulation of Bmal1/Nrf2/antioxidant genes axis occurred at both translational and transcriptional levels with advancing age. Taken together, our data indicated an opposite relationship between the intracellular ROS and expression of Bmal1 and Nrf2-targeted antioxidant genes in aging lenses/LECs.

### 3.2. Aging/Aged C57BL/6 Mouse Lenses and hLECs Showed Significant Loss of Antioxidant Activities

With advancing age, mouse and human lenses/LECs displayed reduced expression of antioxidant defense genes at protein as well as mRNA levels, but it was not clear from the above expression experiments whether antioxidants also lost their enzymatic activities. Previously, our group reported that *Prdx6*^−/−^-deficient LECs, a model for aging, had reduced PLA_2_ and GSH peroxidase activities in comparison to WT Prdx6 (Prdx6*^+/+^*) [75]. Furthermore, reduced Catalase and SOD expression and activities have been observed during human aging [5,21,65,81,86]. However, the fate of the antioxidant enzymes’ activities during aging, which is responsible to maintain redox cellular signaling and their connection to ROS homeostasis, are not systematically elucidated. In this study, we examined the levels of the enzyme’s activities of major Bmal1-Nrf2-mediated antioxidant enzymes in aging mouse lenses and primary hLECs isolated from lenses of subjects of variable ages as shown in Figure 2. To protect the cells against various oxidative stressors, the threshold level of enzyme activities is essential. A quantitation of enzymatic activities (as described in Materials and Methods) revealed a significant decline in Prdx6′s enzymatic activities such as PLA_2_ and GSH peroxidase, Catalase and SOD in an age-dependent fashion as shown in Figure 2, suggesting that not the only expression of antioxidants is decreased, but also their enzymatic activities are significantly diminished. Taken together, data analyses demonstrated that mRNA or protein and enzymatic activities of examined Phase II antioxidants declined with aging, and the loss was more significant in aged samples (Figure 2A–H). As expected, data revealed a significant inverse link between levels of enzymatic activities of examined antioxidants and increased ROS accumulation during aging. These above experimentations provided the base for this study to evaluate the efficacy and the mechanism of action of Metformin in reactivation and restoration of dysregulated Bmal1/Nrf2/antioxidant pathway and cellular protection.

### 3.3. Cell Survival Assays Revealed That 1 mM of Metformin Promoted LECs Growth and Maintained Their Health

To examine Metformin’s effect on lenses or LECs, at first, we intended to determine an effective noncytotoxic concentration of Metformin. Toward this, mLECs or hLECs were exposed to different concentrations of Metformin as shown in Figure 3. Data revealed that 1 mM of Metformin concentration has a noncytotoxic and better effect on cell health of LECs up to the observation period of 24–72 h; The healthy effect of 1 mM Metformin treatment could be observed on mLECs cell line (Figure 3A), primary mLECs (Figure 3B,C) as well as on Emb-hLECs (Figure 3D) and SRA-hLECs (Figure 3E).

### 3.4. Organic Cation Transporters Present in the LECs and Metformin Enhanced Its Expression Level

Furthermore, the organic cation transporters, OCT1, OCT2 and OCT3 have been demonstrated to be the transporters for the Metformin’s pharmacological action and these transporters are known to have cells/tissues specific expression patterns [87,88,89,90,91]. Thus, we wanted to determine whether lenses/LECs express OCTs receptors, and there are any gender or species-specific effect of Metformin on their expression levels. As shown in Figure 4, mLECs derived from both sexes and human LECs were treated with 1 mM of Metformin concentration for 24 h, and processed for RT-qPCR using OCT1, OCT2 and OCT3 specific probes. We found that all LECs whether they are derived from mice or humans were enriched with the OCTs (Figure 4). Next, we were interested to know whether Metformin treatment affects the levels of OCTs expression. To achieve these data, Metformin-treated mLECs of both sexes, Emb-hLECs and SRA-hLECs were subjected to RT-qPCR. Unexpectedly, we observed that basal transcription of all three OCTs was dramatically increased in Metformin-treated LECs whether the LECs were derived from mouse lenses of either sex (Figure 4A–C) or human lenses (Figure 4D–F). However, we did not observe any detectable differences in the expression levels of OCTs in mLECs-derived from either gender, suggesting both sexes of mLECs were equally responsive to Metformin treatment, at least regarding the OCTs transcription (Figure 4). Interestingly, we could notice that expression of OCT1 and OCT3 levels were more significantly increased in both, mouse and human LECs in comparison to OCT2. However, the reason for the increased expression of OCT1 and OCT3 could be due to the abundance of transcription factors responsible for their transcriptional activation. We believe that as whole our data indicate that 1 mM of Metformin has the ability to enhance its own receptors, that in turn makes Metformin more accessible to cells/tissue, and that may facilitate Metformin to produce better cell health benefits. Because our data revealed that 1 mM of Metformin is a noncytotoxic effective dose for better growth and survival for both mouse and human LECs, we used this concentration of Metformin throughout our study unless otherwise stated.

### 3.5. Metformin-Induced Increased Nrf2 Expression Required AMPK Activation as Evidenced by Compound C(CC), an Inhibitor of AMPK, in Human and/or Mouse LECs

It has been reported that Metformin exerts its beneficial effects via AMPK activation-mediated induction of Nrf2/ARE antioxidant pathway, resulting in halt or delay of various age-related diseases [51,92,93,94,95]. Because Metformin has a multifaceted mechanism of action and effects through AMPK-dependent and -independent pathways [96,97], and shows cell/tissue-specific functions [54,98,99,100,101,102], we intended to confirm whether Metformin activates Nrf2, a master transcriptional regulator of the antioxidant pathway, by AMPK pathway in mLECs and SRA-hLECs as reported in other cell types. To examine this, we treated the LECs with Metformin in the absence or presence of AMPK inhibitor, compound C (CC), as shown in Figure 5A,B. The data demonstrated an increased phosphorylated form of AMPK (Figure 5Aa,Ba) in LECs subjected to Metformin treatment, while in presence of CC, Metformin failed to activate the AMPK pathway. Next, we analyzed the Nrf2 expression level in LECs treated with Metformin in the absence or presence of AMPK inhibitor CC as shown in Figure 5Ac,Bc. We observed that Metformin activated the expression levels of Nrf2 protein in untreated LECs, but Metformin failed to activate Nrf2 expression in LECs in the presence of an AMPK-inhibitor. Taken together, results revealed that Metformin boosted the level of Nrf2 expression via AMPK-activation pathways in lens cells (Figure 5) as reported earlier in other cell types [37,38,41]. It would be worth mentioning that Metformin is an activator of the AMPK pathway, and AMPK activation derepresses the Bmal1 by disrupting the stabilization of CRY protein, resulting in Bmal1 activation potential, Since the Nrf2 gene contains functional sites of Bmal1 and Nrf2 expression can also be regulated by its autoregulatory mechanism, we surmised that increased expression of Nrf2 by Metformin should be associated with cooperative Bmal1 and Nrf2 regulation of Nrf2.

### 3.6. Metformin Treatment Enhanced Antioxidant Genes Transcription via Activation of Nrf2 in General

Previously we have reported that Nrf2/ARE-mediated increased transcription leads to enhanced antioxidant genes expression with increased cytoprotection against various kinds of oxidative stress [5,21,81]. To clarify and support our hypothesis that Metformin’s AMPK-mediated reinforced Nrf2 expression is functionally active in LECs as observed (Figure 5) and can reactivate Nrf2/ARE-mediated transcription of all antioxidant genes containing ARE sites, we utilized pRBGP2-3x ARE-LUC containing three ARE sites or its mutant at ARE sites, pRBGP4-3Xmut ARE-LUC [79], (a kind gift from Hozumi Motohashi, Japan). As shown in Figure 6, SRA-hLECs as well as different ages of primary hLECs were transfected with the above-mentioned plasmids along with *Renilla*, pRL-TK vector (Promega) as shown in Figure 6A,B. Transfectants treated with Metformin for 24 h were assayed for luciferase activity using Dual-Glo luciferase assay system with a 96-well plate (Promega, Madison, WI, USA) and normalized with *Renilla* O.D. We observed that Metformin treatment increased the Nrf2/ARE- mediated transcription in a dose-dependent manner in SRA-hLECs as shown in Figure 6A. Interestingly, the data derived from aging LECs suggested that Nrf2/ARE-mediated antioxidant pathways were functionally dysregulated in hLECs with advancing age and that could be reactivated by Metformin through Nrf2/ARE activation (Figure 6B). The findings also provided a clue to further design experiments to delineate the molecular mechanism-based elucidation of Metformin’s activity in the regulation of antioxidant genes containing ARE, like Prdx6 and protection of LECs.

### 3.7. Metformin Treatment Augmented Bmal1, Nrf2, and Nrf2/ARE Antioxidant Genes Expression and Amplified the Enzymatic Activities in mLECs

In the previous experiments of this study, we found that Metformin increased the Nrf2 expression levels by activation of AMPK pathways. In our published report, we have shown that Nrf2 and Bmal1 cooperatively regulate the antioxidant gene expression such as Prdx6 [21]. Metformin acts through the activation of AMPK. It has been shown that AMPK activation pathway derepresses Bmal1 and thereby revives transcription of its target genes, by destabilizing the CRY’s cellular activity [31]. Here, we intended to know whether Metformin upregulated the Bmal1 expression and its upregulation could be connected to the upregulation of Nrf2 and antioxidant genes since functional Bmal1/E-Box sites are present in Nrf2 and Prdx6 gene promoters [21]. Total protein and RNA were isolated from Metformin-treated mLECs and subjected to expression assays, RT-qPCR and Western analyses. Expression assays showed a significantly increased expression of Bmal1, Nrf2 and Prdx6 mRNA (Figure 7A–C) as well as protein (Figure 7D) in response to Metformin, suggesting that Nrf2 and Prdx6 expression were increased via Metformin-mediated AMPK activation-driven Bmal1 and Nrf2 expression and activation. However, we would not be able to surmise how and what transcription factor(s) are involved in the upregulation of Bmal1 expression in LECs. We think that a separate line of research will be required to delineate Metformin-mediated transregulation of Bmal1 expression. Moreover, next we wished to analyze Metformin’s effect on enzymatic activities in Metformin-treated LECs, such as Phospholipase A_2_ (Figure 7E), GSH peroxidase (Figure 7F), Catalase (Figure 7G) and SOD (Figure 7H). Figure 7 showed a significant increase in enzymatic activities in Metformin-treated mLECs. We reasoned that Metformin acts as an anti-aging molecule and Metformin exposed cells will be physiologically healthier and, therefore, the antioxidant’s enzymatic activities will be more responsive. However, we do not know the exact mechanism of how Metformin regulated the enzymatic activities of antioxidants. Furthermore, we surmise that Metformin may alter the configuration of the antioxidant enzyme by direct binding and thereby stabilizing them to gain active status within the cellular microenvironment. However, our data revealed that Metformin-treated mLECs showed a significantly higher expression of Bmal1, Nrf2, with increased antioxidant gene expression and activities as shown in Figure 7.

### 3.8. Metformin Mitigated the ROS Levels by Upregulating Bmal1, Nrf2 and Antioxidant Genes Expression in Different Ages of Primary mLECs Isolated from Male or Female of C57BL/6 Mice

Based on the above experiments, we sought to determine whether Metformin can induce the basal gene expression levels in primary mLECs isolated from aging or aged mice of both sexes. However, due to less availability of primary aging mLECs, we have performed limited experiments in these cells. Our experimentation on aging LECs revealed that aging lenses/LECs displayed higher levels of ROS (Figure 1A), and the increased accumulation of ROS-induced oxidative stress is an established cause for age-related etiopathologies. Metformin bears anti-aging properties and can attenuate aging-associated oxidative stress-induced injuries by augmenting antioxidant responses (as stated in the Introduction section). Thus, we wanted to know whether Metformin is efficacious in mitigating ROS levels in primary aging/aged mLECs as previously reported in other cell types [36,37,38,39,40,41]. Primary mLECs isolated from C57BL/6 male or female mice of variable ages were treated with 1mM of Metformin for 24 h. We found that Metformin significantly lessened the levels of ROS in mLECs-derived from lenses of 15 months (M) and 21 M old C57BL/6 mice of both genders as shown in Figure 8A. Human age equivalents to 15 M and 21 M old mice are approximately 50 and 62 years. To examine the expression levels of antioxidant response, another set of Metformin-treated primary mLECs were processed for RT-qPCR. The data revealed that Metformin-treated primary mLECs of both genders had a significant increase in Bmal1 (Figure 8B), Nrf2 (Figure 8C) and antioxidant genes, such as Prdx6 (Figure 8D), Catalase (Figure 8E) and SOD1 (Figure 8F) mRNA expression as shown in Figure 8. Collectively, data demonstrated that Metformin augmented antioxidant gene transcription (mRNA) by upregulation of Bmal1 and Nrf2 activation.

### 3.9. Metformin Induced Bmal1 and Nrf2-Dependent Antioxidant Gene Transcription in Emb-hLECs

Antioxidant gene response can be different in different cell types of different genetic backgrounds, and also cells derived from different ages can respond differentially to Metformin treatment. Metformin is prescribed to treat subjects of variable ages. Thus, we next examined whether the results obtained from mLECs or adult/aged hLECs were reproducible and whether Metformin treatment is equally effective in inducing antioxidant pathways in the primary culture of Emb-hLECs. Toward this, we examined the efficacy of Metformin in inducing the expression status of Bmal1, Nrf2, and Nrf2 antioxidant genes such as Prdx6, Catalase and SOD1. Metformin-treated Emb-hLECs were processed for RT-qPCR, protein expression and antioxidant enzymatic activities analyses as stated in the Materials and Methods section. We observed that basal transcription of these genes was dramatically increased in Metformin-treated Emb-hLECs, as evidenced by increased mRNA (Figure 9A–E; orange vs. blue bar), suggesting that these cells are almost equally transcriptionally responsive as observed with other LECs. Furthermore, in another set of experiments, Metformin-treated Emb-hLECs were submitted for protein expression analyses. An equal amount of total cell lysates was immunoblotted with antibodies specific to Bmal1 or Nrf2 or Prdx6. Results revealed a significant increase in the expression of these three proteins (Figure 9F). Next, to measure the effect of Metformin on enzymatic activities of antioxidants, the same protein lysates were used that were used for protein expression analyses. As expected, Metformin treatment significantly increased the enzymatic activities such as PLA_2_ (Figure 9G), GSH peroxidase (Figure 9H), Catalase (Figure 9I) and SOD (Figure 9J) in Emb-hLECs. Collectively, the results illustrate that Metformin is equally efficacious in activating antioxidant pathways in LECs whether the LECs are derived from lenses of aged or younger subjects. Furthermore, results support that Metformin can be used to block or delay adverse signaling without discriminating against aging status.

### 3.10. Metformin-Dependent Increased Expression of Bmal1 and Nrf2 Was Linked to Increased Antioxidants Expression in SRA-hLECs

Next, we sought to elucidate whether Metformin stimulates the basal level of antioxidant gene expression and activities in the human lens epithelial cell line, SRA-hLECs, a cell line, which is profoundly used to investigate the effects and the molecular mechanisms of biomolecules/compounds of endogenous or exogeneous origin, as primary LECs scarce and prohibitively expensive. Interestingly, similar to primary LECs, we observed that SRA-hLECs treated with Metformin for 24 h showed increased expression of all five genes, Bmal1, Nrf2, Prdx6, Catalase and SOD1 (Figure 10A–E) as observed with Emb-hLECs or mLECs. Next, we examined the protein expression of Bmal1, Nrf2 and Prdx6 protein in SRA-hLECs treated with Metformin. Western analysis with their corresponding specific antibodies showed increased protein expression (Figure 10F). Further, the enzymatic activities estimation of antioxidants demonstrated that the activities of Phospholipase A_2_ (Figure 10G), GSH peroxidase (Figure 10H), Catalase (Figure 10I) and SOD (Figure 10J) were increased in the same protein lysate used for the Western analysis (Figure 10F). As a whole, the outcomes argued that SRA-hLECs are equally valuable reagents as primary cells since the data obtained were well comparable to data derived from primary LECs.

We next identified the effect of Metformin on the fate of the Nrf2 in SRA-LECs. To achieve this, we assessed the Nrf2 protein level in cytosolic and nuclear extracts of SRA-hLECs treated with different concentrations of Metformin for 24 h. Immunoblot data obtained by using an antibody specific to Nrf2 revealed that Nrf2 migrated at approximately 110 kDa on SDS-PAGE and was enriched in nuclear extract of SRA-hLECs. Maximum levels of nuclear Nrf2 could be observed at 1mM of Metformin concentration as shown in Figure 10K. While cytosolic extract had a residual protein amount of Nrf2. However, we also detected the nuclear Nrf2 in the untreated control, SRA-hLECs, suggesting that a low level of Nrf2 in the nuclear fraction of LECs may be essential for basal expression and activity of Nrf2 to maintain cellular physiology. Moreover, we could notice a dramatic increase of nuclear Nrf2 in Metformin-treated SRA-hLECs, we posited that Metformin may increase the DNA binding/activity of Nrf2 by increasing its abundance (Nrf2′s), and AMPK-mediated Nrf2 activation (phosphorylation) [41] and Bmal1-mediated increased transcription [31]. Next, we asked whether Metformin induced an increased cellular abundance of Nrf2 is functionally active, to investigate this possibility, we examined the Nrf2 transactivation capacity as described in Materials and Methods. Nuclear extract isolated from SRA-hLECs treated with different concentrations of Metformin, having an equal amount of protein samples was used to quantify the Nrf2 activity with Trans-Nrf2 transcription factor assay (active motif) as shown in Figure 10L. The data from experimentation demonstrated that Nrf2-DNA binding activity progressively increased with increasing concentrations of Metformin, suggesting that Metformin has the potential to increase the DNA-Nrf2 interaction.

### 3.11. Metformin Treatment Revived Bmal1, Nrf2 and Prdx6 Gene Expression and Mitigated the Elevated ROS Levels in Aging hLECs

In our published report, we showed a progressive decline in antioxidant gene expression and increased ROS generation during aging [21]. Furthermore, in this study, our results revealed that increased ROS accumulation in lenses of mice was related to reduced expression of antioxidant defense gene expression during aging as shown in Figure 3, Figure 4, Figure 5, Figure 6, Figure 7, Figure 8, Figure 9 and Figure 10. Next, we performed the experiments on whether Metformin has efficacy to mitigate ROS accumulation and can reactivate Bmal1/Nrf2-mediated antioxidant response in primary hLECs isolated from lenses of variable ages of both sexes as observed in our previous experimentation with cell lines of human**/**mice LECs, and mouse aging lenses. To achieve this objective, hLECs were isolated from the lenses of subjects of different age groups of both genders and treated with Metformin. Expression assay conducted using RT-qPCR demonstrated that Metformin could reinforce the expression of Bmal1 and Nrf2, and their target antioxidants genes, such as Prdx6, in aging/aged hLECs of both genders as evidenced by a significant increase in mRNA expression (Figure 11). Furthermore, the Quantitation of ROS levels using H2-DCF-DA dye demonstrated that the levels of ROS could be dramatically decreased in Metformin-treated hLECs (Figure 11A). However, we noticed that expression levels of the above-mentioned molecules reactivated by Metformin were higher in younger hLECs compared to aging**/**aged hLECs but were significantly reactivated in aging/aged hLECs compared to untreated control (Figure 11, orange vs. blue bars), suggesting that younger hLECs were relatively more responsive to Metformin treatment. The results further emphasized that the aging hLECs retained antioxidant pathways and can be reactivated by Metformin to block the age-related progression of ROS generation-driven cellular damages. Conclusively, the results revealed that Metformin is efficaciously effective to boost the antioxidant pathway by upregulating the expression of Bmal1/Nrf2/antioxidant genes significantly, in aging/aged cells of both sexes. The findings also support that at least, one of Metformin’s anti-aging effects occurs via reactivation of antioxidant pathway in aging primary LECs of both genders.

### 3.12. Metformin Promoted the Interaction of Bmal1/E-Box and Nrf2/ARE in Aging/Aged hLECs

Because Metformin activated the antioxidant protective pathway, Bmal1/Nrf2/antioxidant gene axis, at transcripts levels (Figure 7, Figure 8, Figure 9, Figure 10 and Figure 11), we surmised that Metformin exerts its beneficial action via reinforcing transcriptional machinery to augment transcription of examined antioxidant genes in aging LECs. Thus, to investigate the molecular mechanism of Metformin-driven increased expression of antioxidant genes and to define the role of Bmal1 and Nrf2 in this context in vivo, we carried out chromatin immunoprecipitation (ChIP) experiments to determine the occupancy of Bmal1 at E-Box and Nrf2 at ARE sequences present in the regulatory region of Prdx6 gene promoter. hLECs of different ages were treated with Metformin for 24 h, and then were processed for ChIP assay with antibodies specific to Bmal1 (Figure 12B) and Nrf2 (Figure 12C), while nonspecific IgG served as a control vehicle (Figure 12D) as described in Materials and Methods. The results of experiments, Figure 12B,C, demonstrated that the *Prdx6* promoter containing E-Box and ARE sequences were specifically enriched with Bmal1 and Nrf2, respectively. No amplicon could be detected with IgG control, indicating the specificity of Bmal1 and Nrf2 antibodies. These data demonstrated that Metformin augmented the occupation of Bmal1 and Nrf2 at their respective binding sites, E-Box and ARE sequences of *Prdx6* promoter, and explained the mechanism of Metformin-dependent increased transcription of antioxidant genes, like Prdx6 transcription. Similar to Prdx6 gene regulation, we believe that Metformin activates other antioxidant genes containing Bmal1 and Nrf2 binding sites in their promoters. Moreover, we observed that younger hLECs were relatively more responsive to Metformin. However, the data revealed that the aging hLECs retained Bmal1 and Nrf2 activity and were significantly responsive to Metformin treatment. Because ChIP assays directly provided the evidence of enrichment of Bmal1 and Nrf2 at E-Box and ARE sites as well as we already have performed experiments on Metformin’s effects on the cellular status of these transcription factors (Figure 7, Figure 8, Figure 9, Figure 10 and Figure 11), we did not perform Western analysis of Metformin-treated hLECs to examine the nuclear and cytosolic levels of Bmal1 or Nrf2.

### 3.13. DNA Binding and Knock Down Studies Demonstrated That Bmal1 or Nrf2 Enrichment at E-Box or ARE Sites Was Linked to Metformin-Induced Increased Cellular Abundance of Bmal1 or Nrf2

The SRA-hLECs treated with Metformin showed enhanced Bmal1, Nrf2 and other antioxidant genes (Figure 10) expression. In addition, the results of Figure 12 revealed the effect of Metformin on Bmal1 and Nrf2 binding activity to E-Box and ARE sites of *Prdx6* promoter. Next, we asked whether Bmal1′s and/or Nrf2′s cellular abundance influences their binding status as their expression levels significantly decrease with aging. To elucidate this, we performed ChIP RT-PCR with Metformin-treated SRA-hLECs. Metformin-treated SRA-hLECs were processed for ChIP assay with *Prdx6* promoter using ChIP grade antibodies specific to Bmal1 (Figure 13B) or Nrf2 (Figure 13C). As shown in Figure 13, E-Box or ARE sequences were selectively occupied by Bmal1 or Nrf2, respectively, and enrichment of Bmal1 to E-Box or Nrf2 to ARE sequences was concentration dependent in the *Prdx6* promoter. We did not observe amplicon with control IgG, indicating the specificity of the Bmal1 and Nrf2 antibodies. Results revealed that increased abundance of Bmal1 and Nrf2 by Metformin significantly enhanced the Bmal1 and Nrf2 availability at E-Box and ARE site, demonstrating that Metformin’s one of the mechanisms of action in reactivating antioxidant genes by elevating levels of Bmal1/E-Box- and Nrf2/ARE- dependent enhanced transcription.

Furthermore, although it was clear from our earlier experimentation of this study and the above experiments that Metformin-induced increased binding of Bmal1 and Nrf2 to their respective responsive elements in the antioxidant gene promoter was associated with increased cellular levels of Bmal1 or Nrf2, we wanted to further confirm whether it is indeed one of the mechanisms of Metformin-mediated cellular defense. To achieve this, we performed ChIP-RT-PCR of *Bmal1*-depleted SRA-hLECs. SRA-hLECs stably infected with lentiviral specific *sh*RNA to Bmal1 or control *sh*RNA were treated with Metformin for 24 h and ChIP assay was conducted with an antibody specific to Bmal1 and IgG control as shown in Figure 13D. We found a significant reduction of Bmal1/E-Box interaction in *Bmal1*-depleted SRA-hLECs in comparison to control, however, we observed an increase of Bmal1/E-Box interaction in Metformin-treated *sh*Control, but not in *Bmal1*-depleted cells as shown in Figure 13D. The PCR product was undetectable in control IgG samples, validating the specificity of the antibody and specific interaction on Bmal1 with E-Box sequences. Similar results were obtained with *Nrf2*-depleted SRA-hLECs (Data not shown). The results suggests that Metformin-mediated increased cellular abundance of transcription factors, Bmal1 and Nrf2 are responsible for a Metformin-mediated increase binding of Bmal1 and Nrf2 to their response elements of antioxidant genes.

### 3.14. Metformin’s Inefficacy in Activation of Mutant Prdx6 Promoter Uncovered That Transactivation Was Predominately Derived from Direct Binding of Both, Bmal1 to E-Box and Nrf2 to ARE, in Prdx6 Gene Promoter In Vivo

To examine whether Metformin-mediated increased DNA-binding, Bmal1/E-Box and Nrf2/ARE are functional, we carried out transactivation assays using aging hLECs and SRA-hLECs coupled with wild-type (WT) or mutant promoter of Prdx6 as illustrated in Figure 14. To this end, we transfected primary hLECs derived from younger (23 y old) and older (55 y and 71 y) subjects, with WT *Prdx6* promoter fused to CAT reporter plasmid having E-Box and ARE sequences along with GFP plasmid. These transfectants were exposed to Metformin for 24 h as shown in Figure 14B and subjected to CAT ELISA analyses. We observed that Prdx6 promoter had a robust increase of CAT activity in response to Metformin (Figure 14B; orange vs. blue bars), suggesting that Metformin upregulated Prdx6 transcription through E-Box and ARE in aging hLECs. The results demonstrated that younger hLECs were more responsive to Metformin treatment than aged cells, suggesting that aging cells could have active transcriptional programs, and could be reactivated by Metformin. (Note: since aging primary LECs are scarce and prohibitory expensive, we performed further experiments on the transactivation potential of Bmal1 and Nrf2 using SRA-hLECs). SRA-hLECs have been found to be a wonderful reagent for exploring the effects of drug and their mechanisms. Thus, next, we sought to determine the contribution of each site(s), Bmal1/E-Box and/or Nrf2/ARE elements, in Metformin-mediated activation of antioxidant gene, Prdx6 transcription. We utilized Prdx6 gene promoter linked to CAT plasmid bearing Bmal1 and Nrf2 response elements or mutant promoter mutated at E-Box or ARE site or both sites for experimentation. SRA-hLECs were transfected with wild type (WT) promoter plasmid-linked to CAT or its mutants, Bmal1 specific mutant at E-Box (E-Box-mut), or Nrf2 specific mutant at ARE (ARE-mut) or at both Bmal1/E-Box and Nrf2/ARE (E-Box-mut + ARE-mut) promoter constructs linked to CAT reporter plasmid along with GFP vector as shown in Figure 14A. After 48 h, transfectants were treated with Metformin for 24 h as shown in Figure 14C. We observed that mutation at E-Box showed ~75% reduction and mutation at ARE site showed ~60% reduction in the promoter activity. However, the promoter having a mutation at both sites displayed only ~5–7% transcriptional activity. Transfection efficiency was equalized with GFP O.D. Furthermore, we found that transfectants treated with Metformin showed a significant increase in the activity of WT promoter, while the mutant promoter(s) at single site E-Box or ARE or mutant at both sites E-Box and ARE showed significantly less response to Metformin treatment, indicating that both sites, E-Box and ARE, are essential elements for Metformin-mediated transactivation of antioxidant genes transcription.

For further validation of Metformin-mediated increased transcription of antioxidant genes via Bmal1 and Nrf2, we carried out the transient transfection experiments in Bmal1-and Nrf2-depleted SRA-hLECs as noted in Materials and Methods. *Bmal1*-depleted (Figure 14Da) and *Nrf2*-depleted (Figure 14Ea) SRA-hLECs were transfected with WT Prdx6 promoter-fused to CAT reporter plasmid along with GFP. After 48 h of post-transfection, these transfectants were exposed to 1 mM of Metformin for 24 h. The transfectants were harvested and processed for *Prdx6* promoter activity assay using CAT-ELISA. Data showed a significant reduction in transcriptional activity in both Bmal1- (Figure 14D) as well as Nrf2-(Figure 14E) deficient transfectants. Further, we observed that Metformin treatment significantly increased the promoter activity in LVs*h*-Control or *sh*-Control SRA-hLECs. We noticed that very low activity was present in LV*sh*-Bmal1 and *sh*-Nrf2 SRA-hLECs. We believe that the observed low activity was due to the residual presence of Bmal1 or Nrf2 protein in transfectants. Taken together, data indicated that Metformin-mediated Bmal1/E-Box and Nrf2/ARE expression influences the transcriptional activity of the antioxidant gene, Prdx6, and activation of both transcriptional proteins is linked to Metformin-mediated robust transcription of Prdx6.

### 3.15. Knock Down Experiments Revealed That Metformin Treated LECs Engendered Resistance against Oxidative or Aging Stress-Driven Cellular Derangement by Reinforcing Transcriptional Proteins, Bmal1 and Nrf2 Activities

With our major objective(s) of this study toward developing a transcription-based “inductive therapy” to reactivate dormant transcriptional program of antioxidant defense pathway in aging/sick cells, we selected Metformin because it has anti-aging functions [51,52,53] with many beneficial effects in context to promoting cellular health [94,95,103,104]. Moreover, our eyes are continuously exposed to sunlight and environmental pollutants, and these factors are known inducers of oxidative stress-induced cellular insults. Thus, we examined whether Metformin delivery can blunt the cellular damage caused by oxidative stress via upregulation of Bmal1 and Nrf2 antioxidant pathways. To achieve this objective, we employed knockdown strategy and depleted Bmal1 and Nrf2 in SRA-hLECs by using LV *sh*-Bmal1 and *sh-*Nrf2, respectively, with their corresponding controls (Figure 15A,B) as stated in Materials and Methods. Knockdown efficiency of both Bmal1 and Nrf2 *sh*RNA is already examined (Figure 14Da,Ea) and published by our group [21]. These transfectants were subjected to Metformin treatment, and then exposed to H_2_O_2_-induced oxidative stress and measured for cell viability (Figure 15A,B). Cell viability measurement disclosed that Metformin was significantly ineffective in saving SRA-hLECs bearing LV *sh*-Bmal1 (Figure 15A) or *sh*Nrf2 (Figure 15B). Data were normalized with the absorbance of untreated controls. We also measured ROS levels in these transfectants. Quantitation of ROS by H2-DCF-DA dye revealed that Metformin did not mitigate ROS levels significantly (data not shown), demonstrating that Metformin exerted its protection activity largely through expression and activation of transcription factors, Bmal1 and Nrf2, the major regulator for antioxidant defense pathway. Additionally, the finding suggests that plausibly, one of the major protective activities of Metformin occurs via activation of Bmal1/Nrf2**/**antioxidant genes axis. Furthermore, because our earlier experiments of this study demonstrated that Metformin could reactivate deteriorated antioxidant response in aging/aged LECs by reviving Bmal1 and Nrf2 expression in aging/aged hLECs, we intended to know whether Metformin-mediated reactivation of antioxidant transcriptional machinery in aging LECs is functional. To uncover this, we performed cell viability and ROS expression assays of Metformin pretreated aging hLECs of variable age groups facing oxidative stress as shown in Figure 15C,D. Interestingly, results of viability and ROS measurements revealed that Metformin enhanced viability (Figure 15C) of hLECs by mitigating intracellular ROS (Figure 15D) whether they were derived from younger (23 years old) or aged (73 years old) subjects, suggesting that Metformin was effective to defend aging hLECs by reactivating antioxidant transcriptional response, and findings emphasized that Metformin should be considered as one of the promising reagents to correct the dysregulated functionality of antioxidant response, at least in aging eye lenses/LECs.

### 3.16. Metformin Treatment Successfully Internalized into the Lens and Delayed/Prevented Lens Opacity and ROS Generation Induced by H_2_O_2_

Eye lenses/LECs are suggested to be the best biological model to study the aging-related adverse signaling as well as to identify small molecule’s protective effect and the molecular mechanism of action(s) [62,63]. However, in this study, we discovered that LECs are enriched with all three OCTs receptors required for the internalization of Metformin in cells or tissues. Furthermore, we observed that Metformin enhanced the expression of all OCTs (Figure 4). However, it was uncertain whether Metformin can pass through the eye lens capsule and internalize in LECs. To examine this, we performed lenses in organ culture in vitro and treated them with different concentrations of Metformin (1 mM, 3 mM and 5 mM). We identified that 3 mM of Metformin was the most effective concentration to boost Bmal1/Nrf2/antioxidant axis as in Figure 16E. Lenses isolated from 14 months old C57BL/6 mice (equivalent to ~50 years of human age) were cultured in 199 mediums (without serum) as shown in Figure 16. Extract was prepared (100 mg/mL of 90% methanol) from untreated and Metformin-treated lenses, and incubated at −80 for 2 h, vortexed and centrifuged at 4 °C. Supernatant was processed and analyzed by Acquity UPLC coupled with Waters triple-quad Xevo TQS and presented as a histogram (Figure 16A), demonstrating that Metformin can pass through lens capsule and internalized in the eye lens. Next, in another set of experiments, cultured lenses were pretreated or untreated with Metformin for 24 h in 199 medium and exposed to 100 µM of H_2_O_2_ as indicated in Figure 16B following our previously published reports [67,68,105]. We observed that untreated lenses developed H_2_O_2_-induced lens opacity (Figure 16Bb,d), while Metformin pretreated lenses showed significantly less opacity (Figure 16Bc,e). Lenses were photographed at 44 h (Figure 16Ba–c) and at 96 h (Figure 16Bd,e) and the intensity of the lens opacity is presented in the form of a histogram (Figure 16C). Analysis revealed that Metformin-treated lenses had 72% less opacity when examined at 44 h (Figure 16Bb vs. c; 16C, orange vs. blue-orange bars) and 59% at 96 h (Figure 16Bd vs. e; 16C, orange vs. blue-orange bars) compared to H_2_O_2_ alone treated lenses observed at the same time point (44 h and 96 h; Figure 16C, orange bars). In the parallel experiment, lenses pretreated with Metformin were exposed to H_2_O_2_ to measure the ROS levels (Figure 16D) and expression of Bmal1 and Nrf2 and their target antioxidant gene, Prdx6 (Figure 16E). Data analysis revealed a significant increase of ROS in lenses exposed to H_2_O_2_ alone only, while Metformin pretreated lenses showed a significant reduction in ROS accumulation, suggesting that reduction in ROS levels was connected to Metformin-mediated increased expression of Bmal1/Nrf2/antioxidant mRNA (Figure 16E; orange vs. blue-orange bar). Similar results were obtained at protein levels by Western blot analysis (data not shown). Collectively, the results revealed that Metformin passed through a capsule of aging lenses, internalized in LECs/lenses and thereby, delayed/prevented lens opacity by augmenting Bmal1/Nrf2-mediated antioxidant defense response. These results also provide a proof of concept that Metformin should be considered as an ideal reagent to block aging-related cataracts and provide a base for evaluating its effect in cataract prevention/treatment in aging mice models.

## 4. Discussion

Extensive research during the last decades has shown that aging-related diseases are associated with dysregulation of the antioxidant defense pathway and a continuum of increased oxidative stress [60,85,106,107,108,109], resulting in the pathobiology of cells/tissues. The process also leads to array of inflammatory responses and gene dysregulation [4,67,68,76,85,106,108,109,110]. Nrf2 is a key activator of antioxidant genes in response to aging or oxidative stress [111,112,113]. Recently, several studies, including studies from our own group have shown [2,3,5,7,21,75,76,81,114,115] that the loss of Nrf2 and Nrf2 antioxidant genes expression and activity in aging cells lead to an array of oxidative-induced deleterious responses, impaired function and aging pathologies. This deterioration is proposed to be the primary risk factor for ARDs, such as age-related cataracts (ARC) [7]. Recently, it has been shown that core clock transcription factor Bmal1 is not only involved in the regulation of clock-related gene transcription but also is involved in regulating other non-clock genes transcription [14,18,21,116]. Bmal1 deficiency is found to be a cause of Nrf2 repression and increased oxidative load and cellular injuries [117]. Furthermore, Bmal1-driven activation of Nrf2, including its target antioxidant genes in the mouse lung has been documented [17]. The above-illustrated studies, including our own reports (and as described in the Introduction section) provided us a clue in selecting the drug which can activate Bmal1/Nrf2/antioxidant pathway. We found that Metformin having anti-aging activity [51,52,53,118], should be an ideal molecule for our present study. In the present manuscript, using eye lens/LECs and as a model system, we reported for the first time that with advancing age, oxidative stress amplification is linked to loss of cellular defense due to dysregulation of Bmal1/Nrf2/antioxidant pathway at the transcriptional level, and showed the mechanism(s) involved in the adverse processes. We also observed that the antioxidant activities of most of examined phase II enzymes are dramatically decreased with aging (Figure 1 and Figure 2). As expected, results revealed that like the human aging LECs, mLECs also showed a progressive increase of ROS with advancing age, and the aged LECs were more vulnerable to cell death (Figure 1). We observed that the process was directly connected to the loss of Bmal1/Nrf2/antioxidants expression (Figure 1), with a reduction in antioxidant enzyme activities (Figure 2). Nevertheless, previously we have reported the age-associated reduction of antioxidant enzyme expression [21], suggesting that both, the expression and activity of antioxidants are responsible for cellular insults during aging/oxidative stress. Our data are consistent with results obtained by other model systems showing that the dysregulation of Nrf2 and Bmal1 activities are linked to the loss of antioxidant activities during aging [13,17,18,21,119,120,121,122,123]. Indeed, it was intriguing to observe that Metformin application restored the deranged process of an aging mouse or human LECs by reactivation of transcriptional machinery of Bmal1/Nrf2/antioxidant pathway via AMPK activation (Figure 5). Accumulating data derived from studies in multiple model organisms and cell lines of humans or animals have demonstrated that Metformin abates age-related pathologies [51,124,125]. In addition, it has been shown that Metformin administration in mice extends life span and improves the health span [55,104,126,127].

It has been reported that Metformin internalizes in cells or tissue through receptors, OCTs (Organic Cation transporters) [128,129,130]. Notably, we found that all three OCTs, OCT1, OCT2 and OCT3 were expressed in LECs examined. One of the important findings of the present study was to observe that the expression of the OCTs mRNA was significantly increased in the Metformin-treated cells (Figure 4). This result adds a new and novel characteristic to the efficacy of Metformin’s activity, which has physiological relevance as this can aid Metformin to be more accessible to cells/tissues. We also believe that this phenomenon of Metformin might increase its cellular abundance and thereby making Metformin more effective in mitigating adverse redox cellular signaling in response to conditions of aging status and cellular microenvironment. Since the increased expression of OCTs at the mRNA level, we surmised that Metformin could activate transcriptional machinery by activation of transcriptional proteins genetically allotted for OCTs gene transcription. Studies have shown that OCT1, OCT2 or OCT3 promoters are regulated by epigenetic modification, specifically DNA methylation [131,132]. Furthermore, it has been reported that the promoters of OCTs contain the sequence of cognate E-Box (CACGTG) elements [133]. Moreover, Metformin promotes activation and expression of Bmal1 and plays role in epigenetic reprogramming. We postulate that Metformin-mediated increased expression of OCTs mRNA due to their increased transcription by Bmal1 binding to E-Box of OCTs promoters as well as with epigenetic reprogramming. However, all OCTs share and have some overlapping substrate(s), but with variable kinetics [134,135,136]. We believe that further studies are warranted to investigate Metformin-mediated regulation and amplification of OCTs expression, an important line of research.

We found that Metformin enhances the expression of Bmal1/Nrf2/antioxidants via AMPK activation in LECs derived from both humans and mice of both genders (Figure 7, Figure 8, Figure 9, Figure 10 and Figure 11). The expression and activity levels of Bmal1/Nrf2/antioxidant genes in LECs of male or female mice were indistinguishable, suggesting that Metformin’s mechanism of action could be the same in LECs of both sexes (Figure 8 and Figure 11). The results also demonstrated that AMPK activation is a crucial event for Metformin activation of Bmal1/Nrf2-mediated protective pathway. Figure 5 demonstrates that Metformin failed to activate the protective pathway in LECs pretreated with an inhibitor of AMPK, CC [34,46,47,55,137,138], where these cells showed a significant reduction in the expression of each component of the antioxidant pathway (Figure 5). Studies in multiple research fields with different model systems have shown that in some models, Metformin also can act via AMPK-independent mechanisms and can show cell/tissues specific action [39,99,129,139,140]. Nonetheless, we found that in eye lens/LECs whether derived from mice or humans, Metformin acted through AMPK activation pathway to amplify Bmal1/Nrf2-mediated protective antioxidant pathway (Figure 5). AMPK being the master regulator of metabolism, regulates energy at physiological levels during metabolic imbalance and stress. There is now increasing evidence pointing out that AMPK is a redox sensing molecule and can be activated under the cellular accumulation of reactive oxygen species (ROS), which are endogenously produced due to the loss of antioxidant enzymes in organelles. It has been shown that ROS can activate AMPK by enhancing AMP levels [141]. Of note, the therapeutic potential of AMPK activation has been well recognized as it has diversified beneficial effects, from cell survival to cell death, that can be context-dependent. Similar to AMPK, Metformin (an activator of AMPK) has multiple activities, such as it can enhance cell survival and protection as well as can play role in cancer prevention and treatment [32,129]. In the context of redox biology, Metformin can balance redox signaling by regulating the antioxidant pathway. Our results revealed that Metformin activation of the AMPK-mediated Bmal1/Nrf2/antioxidant pathway, at least in the lens, is highly relevant and beneficial to abate oxidative- or aging-induced cellular damage, where Metformin indirectly mitigates ROS by reactivating high levels of antioxidant response through Bmal1/Nrf2-mediated antioxidant gene expression in aging/aged mouse or human LECs (Figure 7, Figure 8, Figure 9, Figure 10 and Figure 11). It has been shown that for the cells/tissues/organ protection, activation of the Nrf2-dependent antioxidant pathway is required for the beneficial effects of Metformin where internalization of Metformin is required.

Moreover, the eye lens is continuously exposed to various environmental pollutants as well as sunlight (UV-B radiation). These environmental stressors are known to produce oxidative stress, and oxidative stress is recognized as a major cause of age-related diseases, including age-related cataracts (ARC) (as illustrated above and the Introduction section). Our results revealed that Metformin can internalize in eye lenses (Figure 15) and could delay/prevent lens opacity facing oxidative stress by augmenting expression of Bmal1/E-Box and Nrf2/ARE target antioxidant genes, like Prdx6 and mitigating ROS levels (Figure 15). As expected, the data revealed that the Metformin-mediated activation status of the entire AMPK/Bmal1/Nrf2/antioxidant pathway in the eye lens is the same as observed in LECs derived from lenses of mice or humans of variable ages. We showed that Metformin activates the antioxidant protective pathway via the AMPK/Bmal1/Nrf2 pathway by upregulating antioxidant genes (Figure 7, Figure 8, Figure 9, Figure 10 and Figure 11).

Our data revealed that Metformin alters the expression of Bmal1/Nrf2 or Nrf2 antioxidants in hLECs at mRNA levels. Indeed, loss-and gain-of-function experiments coupled with DNA-binding and transactivation assays with Bmal1 and Nrf2 demonstrated that Metformin acted by upregulating the transcription of Bmal1, Nrf2 and antioxidant genes. Furthermore, data analyses uncovered that coordinated action of both transcriptional factors, Bmal1 and Nrf2, is required for threshold activation of antioxidant genes expression and cellular protection against oxidative stress. This finding is further supported by the knocking down of either of the transcription factor(s) showing a significant reduction in transcription/expression of antioxidant genes and cell survival (Figure 15). Mechanistically, we observed that Metformin-mediated increased cellular abundance of Bmal1 and Nrf2 and their increased binding to their corresponding sites, E-Box and ARE in antioxidant genes, resulting in enhanced transcription of the examined antioxidant gene, Prdx6. We surmised that the process could be a major phenomenon of increased cellular availability of antioxidant proteins, resulting in ROS suppression and enhanced cell survival (Figure 15). In addition, our results pointed out that Metformin-evoked activation of AMPK is a primary and critical event for the Bmal1/Nrf2 activation of the antioxidant pathway as AMPK inhibitor (compound C) application disrupted the Metformin-mediated increased expressions and activity of Bmal1/Nrf2 and antioxidants, such as Prdx6, at least in lens/LECs. Finally, we observed that Metformin is efficacious to reactivating sick antioxidant response in aging/aged LECs, and it does so by activating the AMPK/Bmal1/Nrf2/antioxidant gene axis, suggesting that the deranged antioxidant pathway is correctable by Metformin application (Figure 13 and Figure 14). Moreover, extensive studies on Metformin’s mode of action have shown that it works through complex 1 inhibition resulting in AMPK activation [28,55,103,104,127,142]. However, the mechanisms of Metformin have been shown to be linked to redox regulation [143,144,145]. Recently it has been found that Metformin modulates cellular redox balance. A recent overwhelming amount of data supports that Metformin blunts oxidative stress and onset of aging-linked pathologies and extends life health span [54,55]. Furthermore, Metformin has been found to alleviate cellular aging by up-regulating antioxidant pathways in various in vitro and in vivo studies [54,55,94,118,146,147]. Studies have shown that Metformin can exert its protective action with different mechanisms in favor of cellular health; it can promote life-health span through mitohormesis involving Prdx2 as well as by regulating microRNA expression and autophagy [54,148,149,150]. While the pleotropic activities and effects of Metformin present challenges in defining its specific mechanisms that are critical for the observed beneficial effects in preventing/delaying age-related pathologies is an uphill task and required further investigation. Nevertheless, our findings demonstrated that Metformin is incredibly beneficial to defend aging/aged LECs against oxidative stress, by reviving dysregulated antioxidant pathways as shown in Figure 17.

## 5. Conclusions

In conclusion, we have shown for the first time, using a variety of LECs derived from mouse and human lenses of variable ages as model systems and molecular analyses, that Metformin can restore deteriorated antioxidant responses and its adverse impact on cells/tissues/organs, that occurs during aging and oxidative stress, by reactivating AMPK/Bmal1/Nrf2/antioxidant axis as shown in Figure 17. Our results provided proof of the concept that Metformin’s one of the anti-aging mechanisms of cellular or organ protection through reactivation of antioxidant pathway, at least in the eye lens. Mechanistically, our data demonstrated that Metformin activation of AMPK was a primary event for threshold reactivation of transcriptional machinery of the antioxidant pathway, Bmal1/E-Box/Nrf2-ARE. Intriguingly, we found that Metformin activated all OCTs, receptors responsible for its penetration in cells or tissues, suggesting that Metformin itself can make its way(s) for better access to LECs/tissues. We believe that our findings will provide direction for the use of Metformin as a therapeutic molecule to treat/delay various age-related pathologies that in turn lead to disease states, like ARC, and also our study paves a way for future research directions to carry out studies in the aging disease model of eye lenses of mice.

## Figures and Tables

**Figure 1 cells-11-03021-f001:**
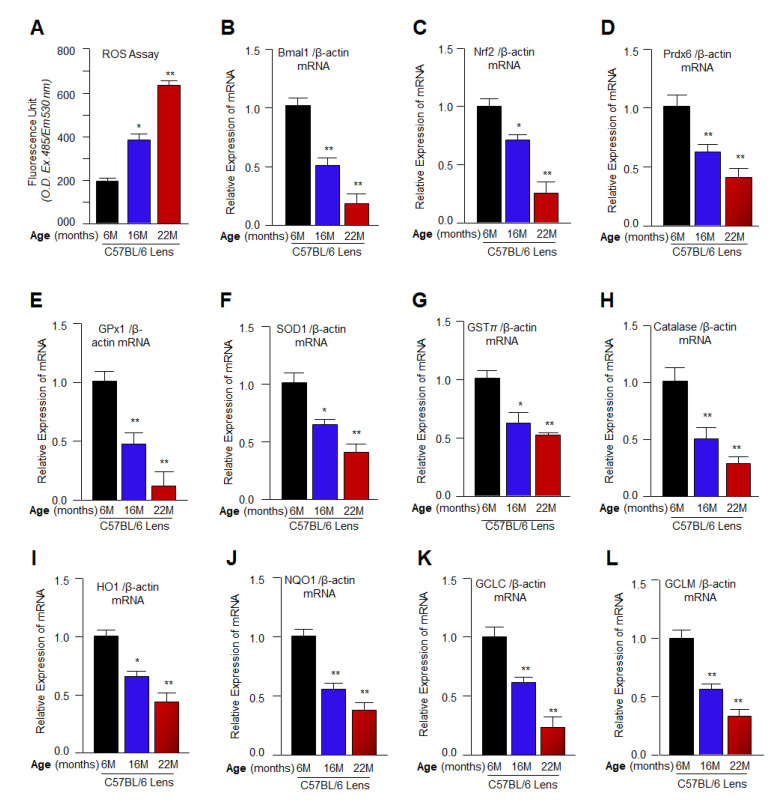
C57BL/6 mouse lenses showed progressive increase of ROS accumulation with a significant decline in Bmal1, Nrf2 and its target antioxidant genes expression during aging. (**A**) Intracellular ROS of lenses of different ages were quantified using H2-DCF-DA dye as described in Materials and Methods and as indicated in figure. Data represents the mean ± S.D. from three independent experiments. Young [6 months (M)] vs. aging (16 M and 22 M) samples. * *p* < 0.05; ** *p* < 0.001. (**B**–**L**) Total RNA was extracted from lenses of different ages and transcribed cDNA was submitted to real-time PCR analysis with specific primers corresponding molecules as indicated. Data represents the mean ± S.D. from three independent experiments. Young (6 M) vs. aging (16 M and 22 M) samples. * *p* < 0.05; ** *p* < 0.001.

**Figure 2 cells-11-03021-f002:**
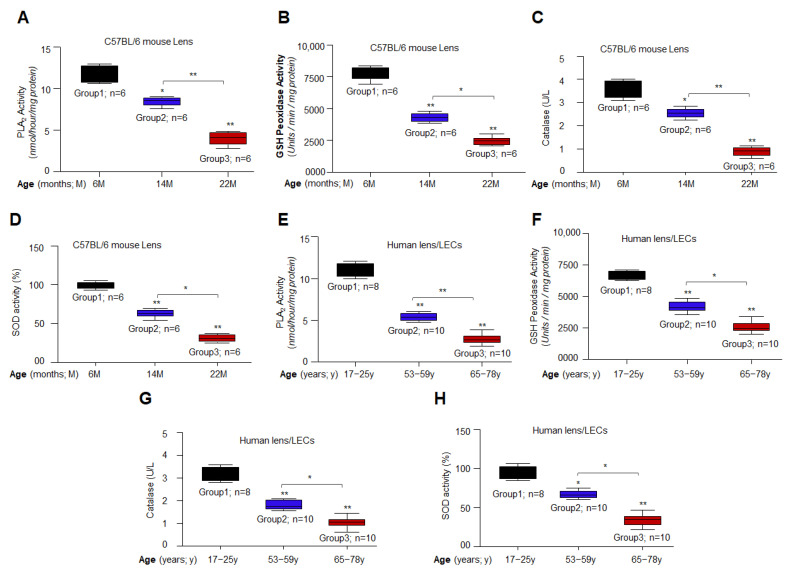
Aging/aged C57BL/6 mouse lenses or hLECs exhibited a significant reduction in phospholipase A2, GSH peroxidase, Catalase and SOD activities. (**A**–**D**) Total protein was isolated from different ages of mouse lenses and processed for analyses of the enzymatic activities. Increasing decline of aiPLA_2_ (**A**), GSH peroxidase (**B**), Catalase (**C**) and SOD (**D**) activities were observed in lenses with aging. Data represents the mean ± S.D. from three independent experiments. Younger (6 M) vs. aging (14 M and 22 M) samples. * *p* < 0.05; ** *p* < 0.001. (**E**–**H**) Primary hLECs isolated from lenses of different ages were divided into three groups: 17–25 y (n = 8); 53–59 y (n = 10) and 65–78 y (n = 10). Total protein was isolated from hLECs of different ages as indicated in figures and was processed for estimating their enzymatic activities. The data represent the mean ± S.D. from three independent experiments. Younger (17–25 y) vs. aging (53–59 y and 65–78 y) samples. * *p* < 0.05; ** *p* < 0.001.

**Figure 3 cells-11-03021-f003:**
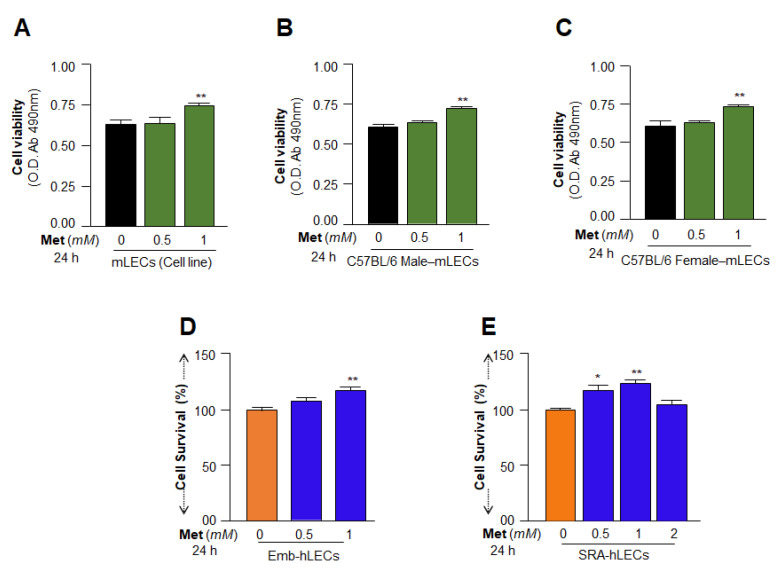
Metformin provided better cell growth of mouse and human LECs. Cell viability assay showing the concentration-dependent effects of Metformin on growth/health of LECs (**A**) mLECs line, primary mLECs isolated from C57BL/6 male (**B**), or female (**C**), Emb-hLECs (**D**) and SRA-hLECs (**E**). Cultured mouse and human LECs were treated with different concentrations of Metformin as shown in figure and were processed for MTS assay to determine nontoxic concentration of Metformin and its effect on cell growth/health. The histograms represent the mean ± S.D. from three independent experiments. Vehicle control vs. Metformin-treated; * *p* < 0.05; ** *p*< 0.001.

**Figure 4 cells-11-03021-f004:**
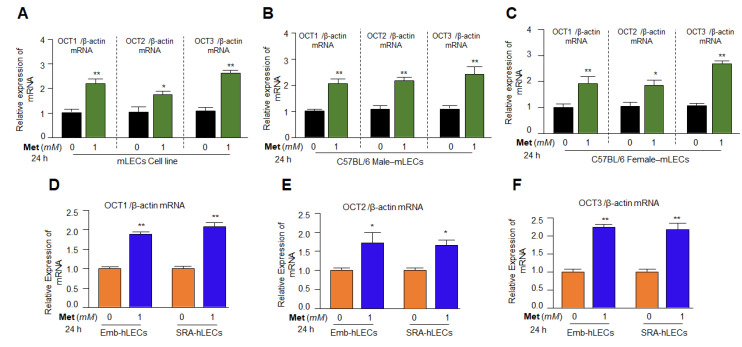
Organic Cation Transporters (OCTs) present in LECs of mouse and human and Metformin upregulated the expression of OCT1, OCT2 and OCT3 in mouse LECs of both sexes and human LECs. mLECs (**A**; 5 × 10^5^/60 mm), primary mLECs (5 × 10^4^/12-well plate) isolated from C57BL/6 male (**B**), or female mouse (**C**), Emb-hLECs (**D**–**F**; 4 × 10^4^/12 well plate) or SRA-hLECs (**D**–**F**; 5 × 10^5^/60 mm) were cultured and treated or untreated with 1 mM of Metformin for 24 h as shown. Total RNA was isolated and subjected for RT-qPCR and Western analyses with specific probes to OCTs. OCT1, OCT2 and OCT3 transcripts were significantly increased in response to Metformin. The histograms represent the mean ± S.D. from four independent experiments. Untreated vs. Metformin-treated; * *p* < 0.05; ** *p* < 0.001.

**Figure 5 cells-11-03021-f005:**
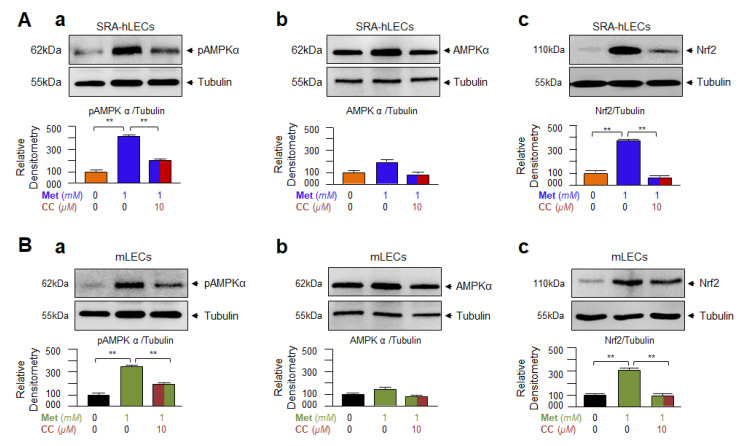
Metformin failed to activate Nrf2 in presence of Compound C(CC), an AMPK inhibitor. SRA-hLECs (**A**) and mLECs (**B**) were culture in 60 mm culture plate overnight, next day cells were treated with an AMPK inhibitor Compound C (CC) and followed by Metformin for 24 h as indicated. Total proteins were isolated and processed for SDS-PAGE immunoblotting with phospho-AMPKα (SRA-hLECs: (**A**,**a**) and mLECs: (**B**,**a**)), AMPKα (SRA-hLECs: (**A**,**b**) and mLECs: (**B**,**b**)) and Nrf2 (SRA-hLECs: (**A**,**c**) and mLECs: (**B**,**c**)) antibodies for protein expression analyses as shown in Figure. Below the protein bands, densitometric analysis of the protein band value normalized with corresponding tubulin and were presented as histograms. Figure (**A**) (SRA-hLECs) and (**B**) (mLECs) show increased phosphorylated form of AMPKα and Nrf2 (second lane in each figure) in response to Metformin treatment. However, Metformin failed to activate the AMPK and Nrf2 in presence of CC, an AMPK inhibitor (last lanes in each figure). Histograms are presented as mean ± S.D. values derived from three independent experiments, ** *p* < 0.001.

**Figure 6 cells-11-03021-f006:**
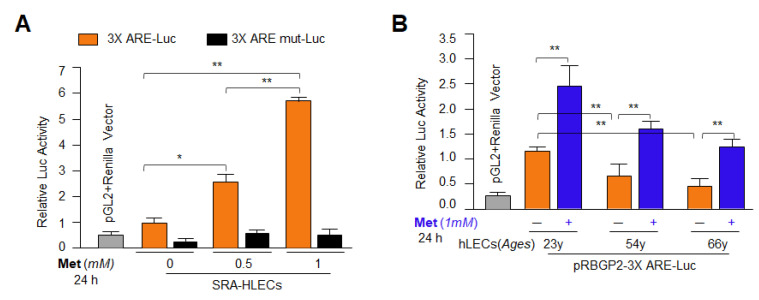
Transactivation assay of engineered promoter containing 3XARE sites suggested that Metformin activated the Nrf2-dependent transcription through ARE in SRA-hLECs (**A**) and even reactivated in aging hLECs (**B**). (**A**) SRA-hLECs were transiently transfected with pRBGP2-3xARE-LUC plasmid or its mutant at all three ARE sites. 12–14 h later cells were treated with different concentrations of Metformin for 24 h as indicated, and relative LUC activity was monitored. All histograms are presented as mean ± S.D. values derived from three independent experiments. Vehicle control vs. Metformin-treated; * *p* < 0.05; ** *p* < 0.001. (**B**) Repression of Nrf2/ARE-mediated transcription in aging was reactivated by 1mM of Metformin treatment. Primary hLECs of variable ages were transiently transfected with pRBGP2-3xARE-LUC plasmid as indicated. 12–14 h after transfection hLECs were washed and treated with control or Metformin for 24 h. Relative LUC activity was monitored. The data represent the mean ± S.D. from two independent experiments. Younger vs. aging samples and control vs. Metformin-treated samples. ** *p* < 0.001.

**Figure 7 cells-11-03021-f007:**
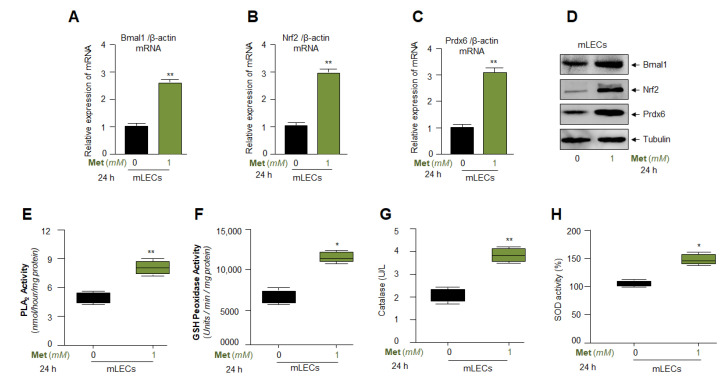
(**A**–**D**) Metformin augmented the expression of Bmal1, Nrf2 and target antioxidant gene, Prdx6 in mouse LECs. mLECs (5 × 10^5^) were cultured in 60 mm culture plate and treated with Metformin for 24 h. Total RNA (**A**–**C**) and protein (**D**) were extracted and submitted to RT-qPCR and Western blot analysis with probes specific to corresponding molecules indicated. Bmal1, Nrf2 and Prdx6 transcripts and protein expression were significantly increased in response to Metformin. The data represent the mean ± S.D. from three independent experiments. Untreated vs. Metformin-treated; ** *p* < 0.001. (**E**–**H**) Metformin amplified Bmal1/Nrf2 target antioxidants enzymatic activities. Equal amounts of protein isolated from mLECs were processed to analyze the enzymatic activities using a commercially available kit. Metformin treatment significantly increased cellular antioxidant enzymatic activities examined; PLA_2_ €, GSH peroxidase (**F**), Catalase (**G**) and SOD (**H**). The data represent the mean ± S.D. from three independent experiments. Untreated vs. Metformin-treated; * *p* < 0.05; ** *p* < 0.001.

**Figure 8 cells-11-03021-f008:**
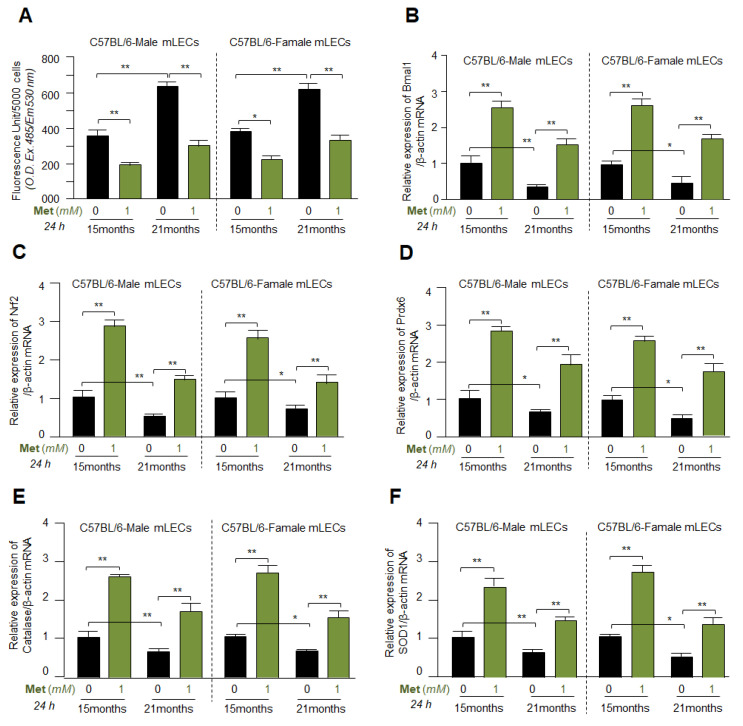
Primary mLECs isolated from C57BL/6 male or female mice revealed significantly reduced ROS level with increased expression of Bmal1 and Nrf2 and target antioxidants, Prdx6, Catalase and SOD1 in response to Metformin. Primary mLECs (5 × 10^4^) isolated from C57BL/6 male and female mice were cultured in 12-well culture plate and treated with Metformin for 24 h as shown in figure. (**A**) Increased ROS levels during aging was mitigated by Metformin treatment. ROS levels were measured using H2-DCF-DA dye methods. The histograms represent the mean ± S.D. from three independent experiments. Untreated vs. Metformin-treated; * *p* < 0.05; ** *p* < 0.001. (**B**–**F**) Total RNA isolated from Metformin-treated primary mLECs-derived from C57BL/6 mice of both sexes was subjected for mRNA analysis of Bmal1 and Nrf2, and their target antioxidant gene, Prdx6, Catalase and SOD1 with their specific primers as shown in Table 1. The data represent the mean ± S.D. from three independent experiments. Untreated vs. Metformin-treated; * *p* < 0.05; ** *p* < 0.001.

**Figure 9 cells-11-03021-f009:**
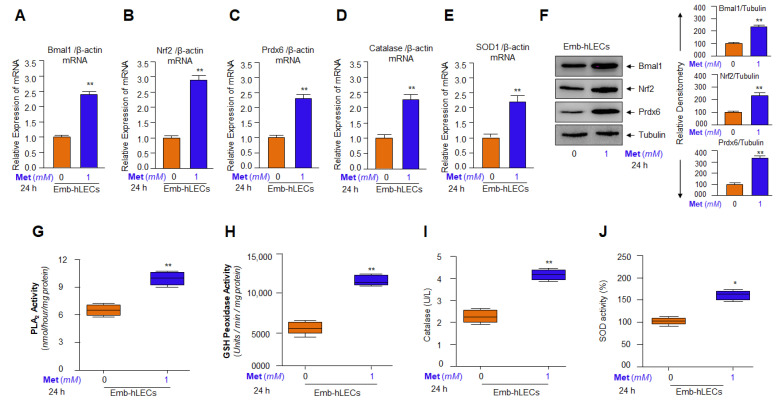
Metformin enhanced the expression of Bmal1 and Nrf2, and their target antioxidant genes, phase II enzymes in primary embryonic human lens epithelial cells (Emb-hLECs). (**A**–**E**) Metformin amplified expression of Bmal1, Nrf2 and its target genes, Prdx6, Catalase and SOD1 mRNA in Emb-hLECs. Emb-hLECs (2 × 10^5^) were cultured in 35 mm culture plate and treated with Metformin as indicated. Total RNA was isolated and processed for RT-qPCR analyses using primers specific to above-mentioned molecules. The histograms represent the mean ± S.D. from three independent experiments. Untreated vs. Metformin-treated; ** *p* < 0.001. (**F**) Metformin enhanced the expression of Bmal1, Nrf2 and its antioxidant target protein, such as Prdx6 in Emb-hLECs. Emb-hLECs (2 × 10^5^) were cultured in 35 mm culture plate and treated with Metformin. Total protein was isolated and subjected to Western analyses. Tubulin was used as internal control. Protein bands were quantified using a densitometer, and levels were normalized to corresponding tubulin levels and presented as histograms in the right side of the protein bands. Untreated vs. Metformin-treated; ** *p* < 0.001. (**G**–**J**) Metformin treatment increased the activities of aiPLA_2_, GSH Peroxidase, Catalase and SOD in Emb-hLECs. Emb-hLECs (2 × 10^5^) were cultured in 35 mm culture plate and treated with Metformin as shown in Figure. Total protein was isolated and enzymatic activities of antioxidants were estimated and presented as indicated. The data represent the mean ± S.D. from three independent experiments. Untreated vs. Metformin-treated; * *p* < 0.05; ** *p* < 0.001.

**Figure 10 cells-11-03021-f010:**
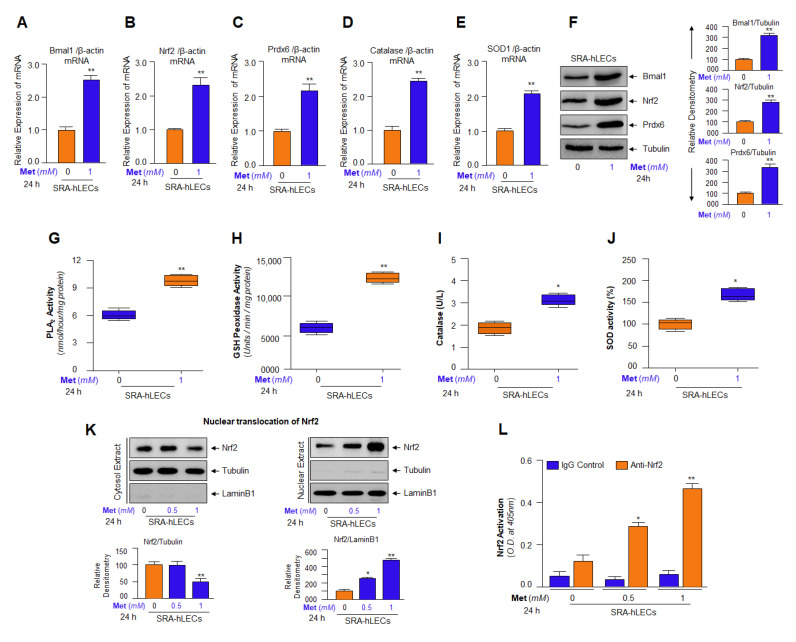
Enhanced Bmal1 and Nrf2-mediated phase II enzymes expression and activities were observed in Metformin-treated SRA-hLECs. (**A**–**E**) Expression of Bmal1 and Nrf2, and target antioxidant genes, Prdx6, Catalase and SOD1 transcripts were increased in Metformin-treated SRA-hLECs. SRA-hLECs (5 × 10^5^) were cultured in 60 mm culture plate and treated with Metformin as indicated. Total RNA was isolated and subjected for mRNA analyses with specific gene primers of the above-noted molecules. Histograms represent the mean ± S.D. from three independent experiments. Untreated vs. Metformin-treated; ** *p* < 0.001. (**F**) Metformin elevated the expression of Bmal1 and Nrf2, and their target genes, such as Prdx6 in SRA-hLECs. SRA-hLECs (5 × 10^5^) were cultured in 60 mm culture plate and treated with 1 mM of Metformin for 24 h. Total protein was isolated and subjected for Western analyses using the specific antibodies of the above-noted molecules. Tubulin was used as a loading control. The histogram shows the densitometric analysis of protein bands which was normalized with corresponding tubulin loading control levels. Untreated vs. Metformin-treated; ** *p* < 0.001. (**G**–**J**) Enhanced activities of aiPLA_2_, GSH Peroxidase, Catalase and SOD were observed in Metformin-treated SRA-hLECs. Total protein was isolated as mentioned above and subjected for enzymatic activities using commercially available kits. The data represent the mean ± S.D. from three independent experiments. Untreated vs. Metformin-treated; * *p* < 0.05; ** *p* < 0.001. (**K**) Increased accumulation of Nrf2 was present in nucleus in LECs treated with Metformin. SRA-hLECs were treated with different concentrations of Metformin as indicated. Cytosolic and nuclear extracts containing an equal amount of protein were used for Western analysis using antibodies specific to Nrf2, Tubulin and laminB1 antibodies. To evaluate the quality of cytoplasmic-nuclear separation, LaminB1 and Tubulin expression were analyzed in both cytosol and nuclear extract. Relative density of the protein bands normalized with corresponding loading control were presented as histograms. Untreated vs. Metformin-treated; * *p* < 0.05, ** *p* < 0.001. (**L**) Transactivation analysis revealed that Nrf2 was upregulated by Metformin. Metformin-treated SRA-hLECs showed a significant increase in Nrf2′s DNA-binding activity to its response element, ARE. SRA-hLECs were treated with different concentrations of Metformin for 24 h, nuclear extracts were analyzed for Nrf2-ARE binding by a commercially available kit. An equal amount of nuclear extracts protein was processed and assayed for Nrf2 activity using anti-Nrf2 (orange bars) or IgG control (blue bars) antibodies according to company’s protocol (Active motif). All histograms are presented as mean ± S.D. values derived from three independent experiments. * *p* < 0.05, ** *p* < 0.001.

**Figure 11 cells-11-03021-f011:**
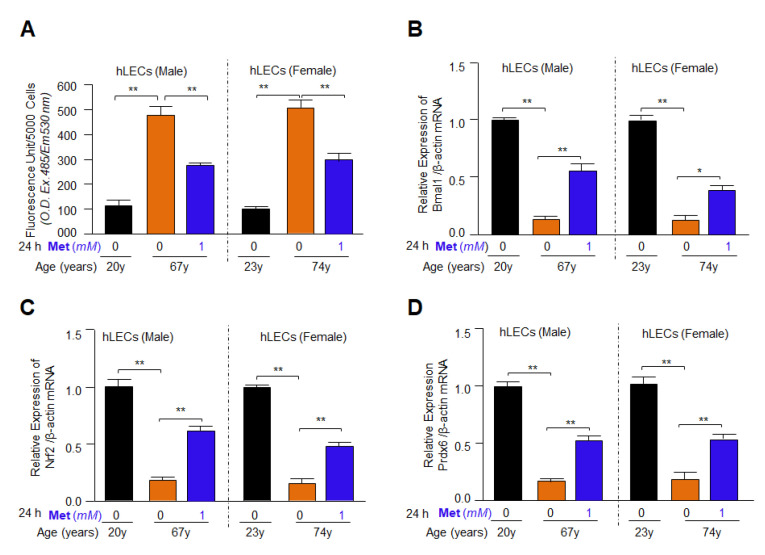
Metformin mitigated age-associated increase of ROS with enhanced expression of Nrf2, Bmal1 and antioxidant genes, such as Prdx6 in aging hLECs of both sexes. Primary hLECs isolated from different ages of male and female subjects were treated with 1mM of Metformin for 24 h. (**A**) Total intracellular ROS levels were measured using H2-DCF DA dye method. All histograms are presented as mean ± S.D. from three independent experiments. Young vs. aging hLECs and Untreated vs. Metformin-treated; * *p* < 0.05; ** *p* < 0.001. (**B**–**D**) Total RNA was isolated and proceeded for RT-qPCR with specific primers of the above-mentioned molecules. Metformin significantly increased the Bmal1 (**B**), Nrf2 (**C**) and antioxidant gene, Prdx6 (**D**) expression levels in aging hLECs of both sexes. The data represent the mean ± S.D. from three independent experiments. Young vs. aging hLECs and Untreated vs. Metformin-treated; * *p* < 0.05; ** *p* < 0.001.

**Figure 12 cells-11-03021-f012:**
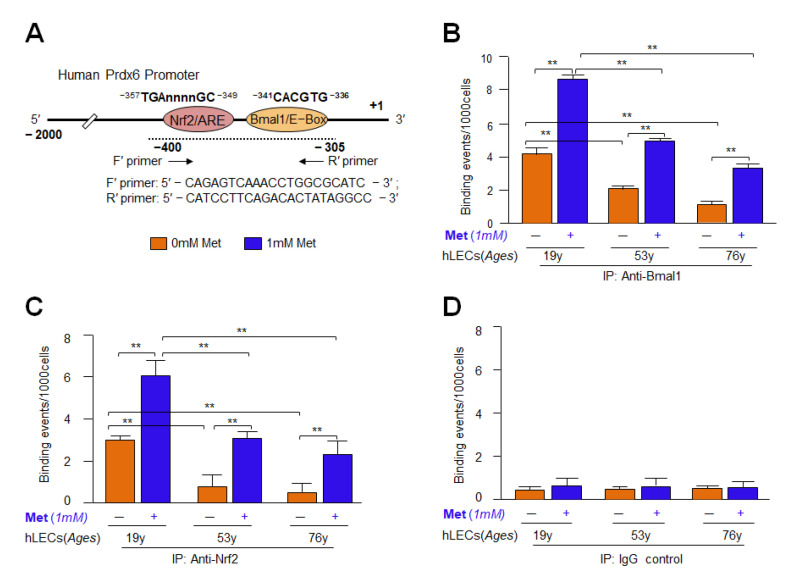
Aging hLECs displayed a significant loss in Bmal1/E-Box or Nrf2/ARE binding and was revived by Metformin. (**A**) Schematic illustration of Prdx6 gene promoter containing Bmal1/E-Box and Nrf2/ARE sites with sequences and position as well as primer location and sequences. (**B**–**D**) In vivo DNA-binding assay, ChIP assay revealed that Metformin revived binding activity of Bmal1 and Nrf2 to their corresponding sites in the *Prdx6* promoter in aging hLECs. ChIP was conducted with antibodies specific to Bmal1 (**B**), Nrf2 (**C**), and nonspecific IgG control (**D**). Immunoprecipitated DNA fragments isolated from different ages of primary hLECs were purified and processed for RT-qPCR analyses using primers shown in Figure. Histograms represent loss of Bmal1 enrichment (**B**; orange bars) at E-Box and Nrf2 (**C**; orange bars) at ARE site, and the sites which was restored by Metformin (**B** and **C**; orange vs. blue bars) treatment as shown. ChIP with IgG control (**D**) antibody was used as negative control. The data represent the mean ± S.D. from three independent experiments. Vehicle control (orange bars) vs. Metformin-treated (blue bars) and young (19 y) vs. aged (53 y and 76 y) subjects; ** *p* < 0.001.

**Figure 13 cells-11-03021-f013:**
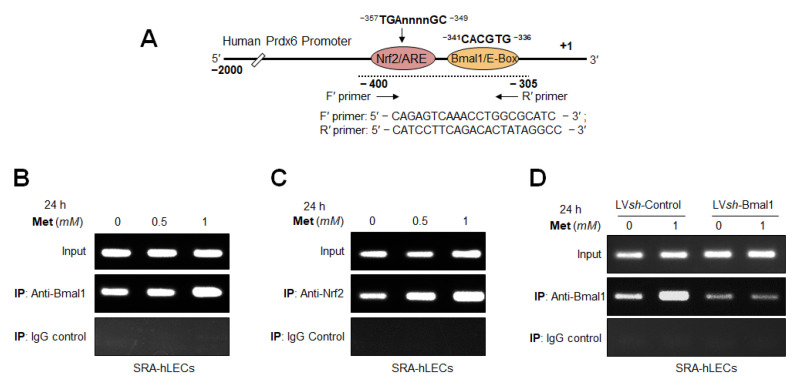
In vivo DNA binding assay revealed enhanced enrichment of Bmal1 to its binding at E-Box and Nrf2 to its ARE sites of *Prdx6* promoter in Metformin-treated SRA-hLECs. Diagrammatic sketch of the 5′-constructs of *Prdx6* promoter containing Bmal1/E-Box- and Nrf2/ARE-DNA binding sites and ChIP primers sequences used for ChIP assay shown in Figure 12A. (**B**,**C**) DNA binding experiments with SRA-hLECs treated with Metformin was performed as noted in Figure 12. Increased binding of Bmal1 and Nrf2 to its responsive elements in the *Prdx6* promoter was observed, suggesting SRA-hLECs responded similarly with Metformin as primary hLECs, suggesting SRA-hLECs are a good alternative for primary hLECs to elucidate the mechanism(s) of protective activity of therapeutic molecules. Chromatin samples were prepared from SRA-hLECs treated with different concentrations of Metformin and were submitted to ChIP assay with ChIP grade antibodies anti-Bmal1 (**B**) and anti-Nrf2 (**C**) and anti-IgG (**B**,**C**). Immunoprecipitated DNA fragments were purified and processed for PCR analysis using primers that specifically recognize fragments of Bmal1 and Nrf2 binding site (−400 to −305) of human Prdx6 gene promoter as indicated. Immunoprecipitated/pulled DNA fragments were subjected to RT-PCR analysis. Fragmented DNA was amplified with primers that specifically recognized a fragment of human *Prdx6* promoter containing E-Box or ARE site. 10% sheared chromatin was used as input. Control IgG was used as negative control. (**D**) Bmal1 knockdown SRA-hLECs involvement of Metformin to enhance the Bmal1 binding to E-Box in the *Prdx6* promoter in vivo. SRA-hLECs were stably infected either with GFP (green fluorescence protein) linked lentiviral (LV) *Sh*-control or GFP-linked LV *Sh* Bmal1 as described materials and Methods section. Genomic DNA isolated from LV *Sh*control and LV *Sh*Bmal1 SRA-hLECs were cross-linked to immobilize bound protein in vivo, was sheared and immunoprecipitated with anti-Bmal1 or control IgG. Immunoprecipitated DNA was amplified using PCR with specific primers. 10% chromatin was used as input DNA. Amplified DNA band visualized with ethidium bromide staining as shown in photographic images.

**Figure 14 cells-11-03021-f014:**
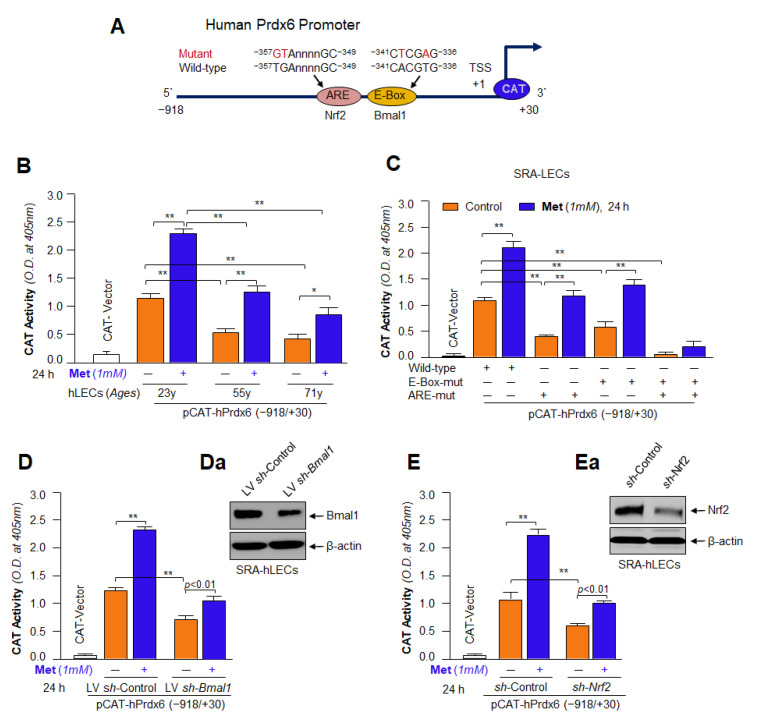
(**A**) Illustration of the 5′- constructs of *Prdx6* promoter (−918/+30 bps) linked to CAT reporter plasmid showing E-Box and ARE site position with wild-type and mutant sequences. (**B**) Age-related reduction in transcriptional activity of *Prdx6* promoter was reactivated by Metformin in hLECs derived from lenses of aging subjects. Histograms showing the *Prdx6* promoter activity in absence or presence of Metformin in different ages of hLECs as indicated. Data represents the mean ± S.D. from two independent experiments. Younger age (23 y) vs. aging sample (55 y and 71 y); untreated vs. Metformin-treated; * *p* < 0.05; ** *p* < 0.001. (**C**–**E**) Transactivation assay with mutant promoters of Prdx6 at E-Box and ARE sites revealed that both Bmal1 and Nrf2 were essential for optimum activation of Prdx6 transcription in SRA-hLECs by Metformin. (**C**) Transcription activation of the wild-type (WT) and mutant (mut) promoters of Prdx6 revealed that both Nrf2 and Bmal1 were critical elements to activate *Prdx6* promoter activity. SRA-hLECs were transfected with WT-pCAT-Prdx6 promoter wild type (WT) plasmid or its mutant promoter plasmids at E-Box (E-Box mut) or ARE (ARE-mut) or at both E-Box and ARE (E-Box-mut + ARE-mut) sites as shown in Figure A. 48 h later, transfectants were treated with vehicle control or Metformin for 24 h as indicated, and Prdx6 transcription activity was measured. All histograms are presented as mean ± S.D. values derived from three independent experiments, ** *p* < 0.001. (**D**) Metformin failed to activate Prdx6 transcription in *Bmal1*-depleted SRA-hLECs. Stably infected SRA-hLECs either with GFP-linked LV control or LV *sh*RNA specific to Bmal1 as described in Materials and Methods and *Bmal1*- knock down efficiency was confirmed by Western blot analysis (**Da**). LV *sh*Control or *Bmal1*-depleted SRA-hLECs were transiently transfected with human Prdx6 promoter (−918/+30) fused with CAT reporter plasmid. 48 h later, the transfectants were treated with Metformin for 24 h as indicated in figure. Cell lysates were prepared and measured the *Prdx6* promoter activity. All histograms are presented as mean ± S.D. values derived from three independent experiments, ** *p* < 0.001. (**E**) *Nrf2*-deficiency in SRA-hLECs resulted in significantly reduced Prdx6 transcription in presence of Metformin. SRA-hLECs were transfected with either mock, negative control *sh*RNA or Nrf2 *sh*RNA. Nrf2 knock down in SRA-hLECs was confirmed by Western blot analysis (**Ea**). The transfectants containing *sh*-control or *sh*-Nrf2- SRA-hLECs were transiently transfected with pCAT-Prdx6 promoter (−918/+30) plasmid. 48 h later these transfectants were treated with Metformin for 24 h and promoter activity was assessed. All histograms are presented as mean ± S.D. values derived from three independent experiments, ** *p* < 0.001.

**Figure 15 cells-11-03021-f015:**
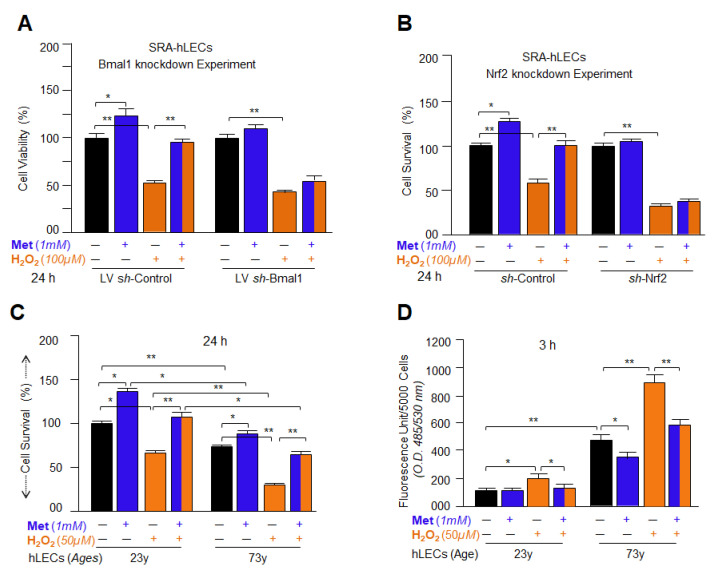
(**A**) Metformin was inefficient to protect *Bmal1-*depleted SRA-hLECs against H_2_O_2_-induced oxidative cell death. Metformin-treated or untreated SRA-hLECs infected with LV *sh*-control or LV *sh*-Bmal1 (Figure 14Da) were exposed to H_2_O_2_ as shown in Figure. MTS assay with Bmal1 knock-down showed that Metformin fails to promote LECs survival against H_2_O_2_ induced oxidative cell damage. All histograms are presented as mean ± S.D. values derived from three independent experiments. * *p* < 0.05; ** *p* < 0.001. (**B**) Similar to Bmal1-knock down experiment (**A**), Nrf2- knockdown experiment demonstrated that Metformin provided cytoprotective activity via Bmal1/Nrf2 protective pathway. Metformin-treated or untreated transfectants containing *sh-*Control or *sh-*Nrf2 SRA-hLECs (Figure 14Ea) were exposed to H_2_O_2_ as indicated in figure. MTS assay with Metformin-treated transfectants containing *sh*RNA specific to Nrf2 showed a significant reduction in cell viability compared to control counterpart. All histograms are presented as mean ± S.D. values derived from three independent experiments. * *p* < 0.05; ** *p* < 0.001. (**C**,**D**) Metformin could be efficacious in defending younger or aged hLECs from H_2_O_2_ induced oxidative stress. Metformin-treated or untreated primary hLECs (4 × 10^3^) were exposed to H_2_O_2_, and Metformin’s effects on cell viability (**C**) and ROS level (**D**) were determined at 24 h and 3 h using MTS and H2-DCF-DA dye assays, respectively and shown. All histograms are presented as the mean ± S.D. from three independent experiments. * *p* < 0.05; ** *p* < 0.001.

**Figure 16 cells-11-03021-f016:**
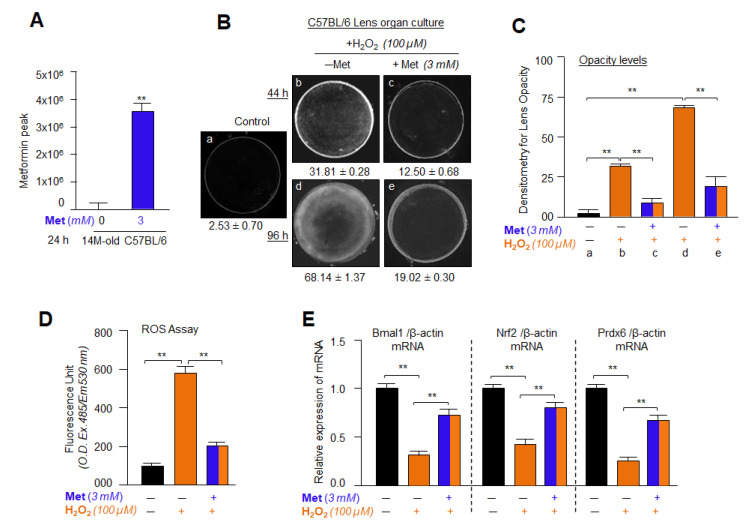
Metformin successfully internalized into mouse eye lenses and delayed/prevented the lens opacity and mitigated ROS accumulation induced by H_2_O_2_ by enhancing the Bmal1/Nrf2/antioxidant, such as Prdx6 in vitro. (**A**) Metformin penetration in the mouse lens cultured in vitro. Lenses isolated from 14 months old mouse were cultured in 48-well culture plate and treated with 3 mM Metformin for 24 h and examined for Metformin penetration using Acquity UPLC coupled with Waters triple-quad Xevo TQS analysis and Metformin peak was presented as histogram. The data represent the mean ± S.D. from three independent experiments. Untreated vs. Metformin-treated; ** *p* < 0.001. (**B**) Lenses isolated from 14 months old C57BL/6 mouse were used for in vitro organ culture as described previously [67,68,85]. The cultured lenses were treated with 3 mM of Metformin followed by 100 µM of H_2_O_2_ exposure. 44 h or 96 h later the lenses were photographed (Nikon SMZ 745T) connected with Camera and software (Nikon). Photograph is representative of three experiments: Metformin significantly prevented the H_2_O_2_-induced lens opacity by 61% and 72% (c and e) compared to H_2_O_2_ (b and d) and untreated lens control (a). (**C**) Histogram indicates densitometry of lens opacity. a, 2.53 ± 0.70 (control, black bar); b, 31.81 ± 0.28 (orange bar: H_2_O_2_ alone, 44 h); c, 12.50 ± 0.68 (blue-orange bar: Metformin + H_2_O_2_ alone, 44 h); d, 68.14 ± 1.37 (orange bar: H_2_O_2_, 96 h); e, 19.02 ± 0.30 (blue-orange bar: Metformin + H_2_O_2_, 96 h). All histograms are presented as mean ± S.D. values-derived from three independent experiments, ** *p* < 0.001. (**D**,**E**) Lenses Isolated from 14 months old C57BL/6 mouse were cultured and treated with 3 mM of Metformin followed by H_2_O_2_ exposure. 48 h later ROS level and mRNA expression of Bmal1, Nrf2 and Prdx6 were analyzed using H2-DCF-DA dye and RT-qPCR analyses, respectively as indicated in Figure, and described in Materials and Methods. All histograms are presented as mean ± S.D. values-derived from three independent experiments, ** *p* < 0.001.

**Figure 17 cells-11-03021-f017:**
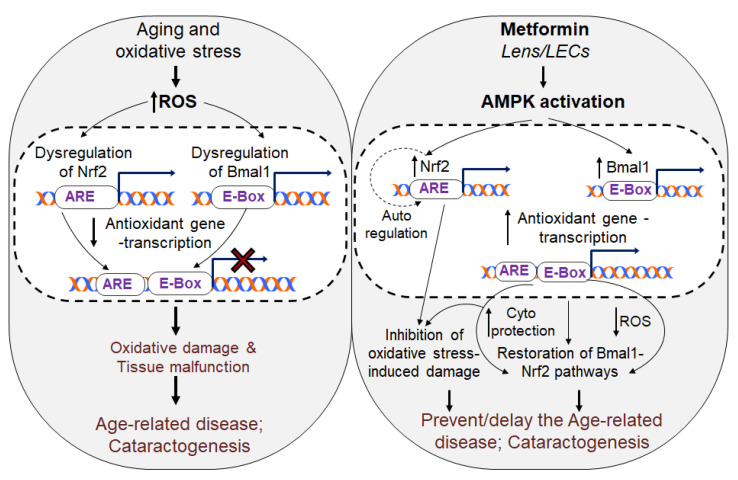
Diagrammatic elucidation of the molecular mechanism of Metformin-mediated restoration of dysregulated Bmal1/Nrf2-dependent antioxidant protective pathway leading to cytoprotection and cataract prevention in response to Aging/oxidative stress. Oxidative stress/aging causes increase in ROS accumulation-driven dysregulation of AMPK/Bmal1/Nrf2 pathways. Our results demonstrated that organic cation transporter (OCTs) present in LECs and upregulated by Metformin treatment. This phenomenon evoked by Metformin can facilitate its increased cellular abundance, and thereby, promoting cellular health through the regulatory interactions of AMPK/Bmal1/Nrf2 and antioxidant genes within the conditions of cellular status and microenvironments during aging and oxidative stresses. We conclude that aging/oxidative stress induced injurious signaling pathways due to deterioration of antioxidant defense mechanism(s) during aging that underlies age-associated diseases; Metformin holds a great promise for combating aging-or oxidative-related pathobiology.

**Table 1 cells-11-03021-t001:** RT-qPCR primers.

Gene	Forward Primer (5′ to 3′)	Reverse Primer (5′ to 3′)
mBmal1	TTTGGGCTAGCTGTGGATAG	AAATATCCACATGGGGGACT
mNrf2	TCTCCTCGCTGGAAAAAGAA	AATGTGCTGGCTGTGCTTTA
mPrdx6	TTCAATAGACAGTGTTGAGGATCA	CGTGGGTGTTTCACCATTG
mGPx1	GTTCTCGGCTTCCCTTGC	GCTGTTCAGGATCTCCTCGT
mSOD1	CAGGACCTCATTTTAATCCTCAC	TGCCCAGGTCTCCAACAT
mGST*π*	TGTCACCCTCATCTACACCAAC	GGACAGCAGGGTCTCAAAAG
mCatalase	CCTTCAAGTTGGTTAATGCAGA	CAAGTTTTTGATGCCCTGGT
mHO1	AGGCTAAGACCGCCTTCCT	TGTGTTCCTCTGTCAGCATCA
mNQO1	AGCGTTCGGTATTACGATCC	AGTACAATCAGGGCTCTTCTCG
mGCLC	AGATGATAGAACACGGGAGGAG	TGATCCTAAAGCGATTGTTCTTC
mGCLM	TGACTCACAATGACCCGAAA	TCAATGTCAGGGATGCTTTCT
mOCT1	TAGCGGCATCAAATCTGGTGGC	CATCTGCAACACAATGGTGGCTC
mOCT2	CGGAGTCTCCAAGATGGTTGATC	CCAGTATCCTCATCTGCCGTCA
mOCT3	CAGCAATGCCTGGATGTTGGAC	TCCTGTGATGCCAACGCCGAAA
mβ-actin	CTAAGGCCAACCGTGAAAAG	ACCAGAGGCATACAGGGACA
hOCT1	CACCCCCTTCATAGTCTTCAG	GCCCAACACCGCAAACAAAAT
hOCT2	GAGATAGTCTGCCTGGTCAATGC	GTAGACCAGGAATGGCGTGATG
hOCT3	CCTTGTCTGTGTCAATGCGTGG	CCAACACCAAGGCAGGATAGCA
hBmal1	GGAAAAATAGGCCGAATGAT	TGAGCCTGGCCTGATAGTAG
hNrf2	TGCTTTATAGCGTGCAAACCTCGC	ATCCATGTCCCTTGACAGCACAGA
hPrdx6	GCATCCGTTTCCACGACT	TGCACACTGGGGTAAAGTCC
hCatalase	CCATCGCAGTTCGGTTCT	GGGTCCCGAACTGTGTCA
hSOD1	TCATCAATTTCGAGCAGAAGG	CAGGCCTTCAGTCAGTCCTTT
hβ-actin	CCAACCGCGAGAAGATGA	CCAGAGGCGTACAGGGATAG

## Data Availability

Not applicable.

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
