# Peer review of "Obligatory Role of AMPK Activation and Antioxidant Defense Pathway in the Regulatory Effects of Metformin on Cellular Protection and Prevention of Lens Opacity"

_cells, 2022, doi:10.3390/cells11193021_

Round 1

Reviewer 1 Report

This is an interesting study in which the authors carried out a comprehensive investigation with large data sets concerning the roles for Metformin in mediating crosstalk pathways involving AMPK and Bmal1/Nrf2/antioxidants and its efficacy in restoring dysregulated antioxidant responses in aging-related pathobiology. Their findings are largely convincing and support their conclusions.

My concerns are as follows:

1) In Figure 10F, the authors showed the induction of Nrf2 by Metformin treatment in SRA-hLECs while in Figure 10K, they showed increased nuclear presence of Nrf2 in cells treated with Metformin.  To include i) Nrf2 in whole cell lysate samples for Figure 10K and ii) imaging analysis of Nrf6 in cells in the absence or presence of Matformin would substantiate their conclusion of Nrf2 nuclear translocation upon the treatment.

2) To include blots showing Bmal1 or Nrf2 knockdown would be necessary for Figures 15A and 15B.

3) On Page 17, the authors stated, “Taken together, results revealed that 637 Metformin boosted the level of Nrf2 expression via AMPK-activation pathways in 638 lens cells (Figure 5) as reported earlier in other cell types.”  It is necessary to include citation(s)/reference(s) about relevant reports in other cell types. Similarly, citation(s)/reference(s) are needed for the authors’ statement on Page 32, “Eye lenses/LECs are suggested to be a best biological model to study the aging- 1209 related adverse signaling as well as to identify small molecule’s protective effect and 1210 the molecular mechanism of action(s).”

4) More proofreading would be helpful to reduce typing errors such as “PhaseII” on Page 1 (Line 18), extra space between “deficiency” and “is” on Page 35 (line 1289).

Author Response

Cells- Manuscript Number:cells-1905259  (“Obligatory role of AMPK activation and antioxidant defense pathway in the regulatory effects of Metformin on cellular protection and prevention of lens opacity” by Chhunchha et al.)

            Firstly, we would like to extend our thanks to the editor and   the reviewers for their careful evaluation of the manuscript, insightful comments and constructive suggestions. In the revised manuscript, we believe that we have addressed the concerns and critiques raised by the reviewers.

            We trust that you will find the issues raised by the reviewers have been adequately addressed and the overall quality of the manuscript has been substantially improved. We now believe it to be acceptable for publication in Cells. Detailed responses to the reviewers’ comments are provided below.

Responses to Reviewer’s Comments

Reviewer 1 (Comments and Suggestions to the Authors):

This is an interesting study in which the authors carried out a comprehensive investigation with large data sets concerning the roles for Metformin in mediating crosstalk pathways involving AMPK and Bmal1/Nrf2/antioxidants and its efficacy in restoring dysregulated antioxidant responses in aging-related pathobiology. Their findings are largely convincing and support their conclusions.

            We thank the reviewer for his/her affirmative comments and suggestions and appreciation of the work.

1) In Figure 10F, the authors showed the induction of Nrf2 by Metformin treatment in SRA-hLECs while in Figure 10K, they showed increased nuclear presence of Nrf2 in cells treated with Metformin.  To include i) Nrf2 in whole cell lysate samples for Figure 10K and ii) imaging analysis of Nrf6 in cells in the absence or presence of Matformin would substantiate their conclusion of Nrf2 nuclear translocation upon the treatment.

We agree, but, with all due respect, we would like to covey that the reviewer noticed (in Fig. 10F) that Metformin treatment augments levels of Nrf2 abundance in whole cell lysates sample compared to untreated control vehicle. However, to avoid repetition of the same data, we did not include the same Western blot figure(s) in Figure 10K. We are certain that the reviewer is fully aware that Metformin mobilizes Nrf2 to accumulate in the nucleus, where it binds and activates antioxidant genes containing ARE sites. Also, it is well established that Metformin activation of AMPK or AMPK activation leads to accumulation of Nrf2 in nucleus (Please see References of this manuscript; 6,36-41, 55, 95, 139). We believe that the reviewer will agree with us as our experimental data reveal that Nrf2 is accumulated in the nucleus after Metformin treatment.

 2) To include blots showing Bmal1 or Nrf2 knockdown would be necessary for Figures 15A and 15B.

Indeed, the reviewer raised a very important point; however, we would like to inform that we used the same transfectants for the experiments of Figure 15 A and 15B, that were used for Figures 14 Da and Ea. Therefore, we did not place these knockdown figures in Figure 15 A or 15B. We have now mentioned this in Figure legends of 15A and 15B to avoid the confusion. Please see Page 31; Lines, 1182,1188-1189.   

3) On Page 17, the authors stated, “Taken together, results revealed that 637 Metformin boosted the level of Nrf2 expression via AMPK-activation pathways in 638 lens cells (Figure 5) as reported earlier in other cell types.”  It is necessary to include citation(s)/reference(s) about relevant reports in other cell types. Similarly, citation(s)/reference(s) are needed for the authors’ statement on Page 32, “Eye lenses/LECs are suggested to be a best biological model to study the aging- 1209 related adverse signaling as well as to identify small molecule’s protective effect and 1210 the molecular mechanism of action(s).”

            We have now incorporated references as suggested by the reviewer.  Please see Page 16; lines 624-626.

4) More proofreading would be helpful to reduce typing errors such as “PhaseII” on Page 1 (Line 18), extra space between “deficiency” and “is” on Page 35 (line 1289).

We have corrected the errors. Also, in accordance with the reviewer’s comments, we have revised the manuscript with special attention to correcting grammatical mistakes or typos. In addition, the manuscript has been reviewed by a professional editor.

Reviewer 2 Report

The manuscript submitted by Chhunchha et al. aimed to investigate how Metformin as an anti-aging drug reduced oxidative stresses to protect lens/LECs. Various in vitro and in vivo mechanistic studies have been performed. They found Metformin reactivated the AMPK/Bmal1/Nrf2/antioxidant defense pathway against oxidative stresses in lens/LECs. Overall, the experiments were well-established and the data were well-organized. The manuscript was clearly written and could be accepted for publishing in Cells if the authors could address the following issues. Thus, I would recommend a minor revision for this paper.

 1.      Page 20, line 770: Metformin successfully internalized into primary mLECs as well as mLECs cell line. But there’s no experiment to directly support this statement. Only the results 3.16. (Page 32-34) showed the internalization of Metformin into the lens.

2.      Page 24, Figure 10K, it should be shown both LaminB1 blot in the cytosol extract and Tubulin blot in the nuclear extract to evaluate the quality of nuclear-cytoplasmic separation.

3.      Page 38, line 1451: “elucidation” should be “elucidates” or “of”.

Author Response

Cells- Manuscript Number:cells-1905259  (“Obligatory role of AMPK activation and antioxidant defense pathway in the regulatory effects of Metformin on cellular protection and prevention of lens opacity” by Chhunchha et al.)

            Firstly, we would like to extend our thanks to the editor and   the reviewers for their careful evaluation of the manuscript, insightful comments, and constructive suggestions. In the revised manuscript, we believe that we have addressed the concerns and critiques raised by the reviewers.

            We trust that you will find that the issues raised by the reviewers have been adequately addressed and the overall quality of the manuscript has been substantially improved. We now believe it to be acceptable for publication in Cells. Detailed responses to the reviewers’ comments are provided below.

Responses to Reviewer’s Comments and suggestions

Reviewer 2 (Comments and Suggestions to the Authors):

…………Overall, the experiments were well-established and the data were well-organized. The manuscript was clearly written and could be accepted for publishing in Cells if the authors could address the following issues. Thus, I would recommend a minor revision for this paper.

We appreciate and thank the reviewer’s affirmative comments and words of appreciation regarding this study.

  1. Page 20, line 770: “Metformin successfully internalized into primary mLECs as well as mLECs cell line”. But there’s no experiment to directly support this statement. Only the results 3.16. (Page 32-34) showed the internalization of Metformin into the lens.

We accept this caveat. Indeed, we did not conduct experiments to show that Metformin internalizes in hLECs. We have now modified the portion by rephrasing the sentences; “collectively, data demonstrated that Metformin augmented antioxidant gene transcription (mRNA) by upregulation of Bmal1 and Nrf2 activation”. Please see page 19; Line 755-757.

  1. Page 24, Figure 10K, it should be shown both LaminB1 blot in the cytosol extract and Tubulin blot in the nuclear extract to evaluate the quality of nuclear-cytoplasmic separation.

Following the reviewer’s suggestion, we have placed LaminB1 as well as Tubulin levels in the figure. Please see Page 23, Figure 10K

  1. Page 38, line 1451: “elucidation” should be “elucidates” or “of”.

            We have corrected the error.  Please see page 37, Line 1438

Reviewer 3 Report

Chhunchha et al. report on “Obligatory role of AMPK activation and antioxidant defense pathway in the regulatory effect of Metformin on cellular protection and lens opacity prevention against aging/oxidative stress”.  

This well-conducted study provides proof that metformin use can restore the antioxidant response mediated by Bmal1-Nrf2-decreased expression and activity during ageing and oxidative stress. The authors also provide evidence that metformin works equally well in LECs from mice and humans of various ages. They discover that metformin's activation of AMPK is a key factor in the expression and activation of the Bmal1-Nrf2-and-tioxidant pathway as well as cytoprotection.

By proving that metformin upregulates its transporters, OCT1, OCT2, and OCT3, in LECs, the work significantly advances the area. Using various the promoter constructs and their mutants, and expression plasmids with expression experiments, they have shown that both, Bmal1 and Nrf2 are needed for antioxidant genes expression and cytoprotection as well as for reactivation of dysregulated antioxidant pathway. Given the physiological significance of this finding in relation to the internalisation of metformin and its effects on cells and tissues, the study is complete. The potential effects of this work on the use of metformin for postponing or preventing age-associated disorders are very significant because they provide very clear evidence for the mechanism(s) by which metformin exerts its protective effects and for how it activates and controls the AMPK-Bmal1-Nrf2-mediated defense response. Therefore, I would like to recommend that the authors discuss this novel finding by referring lens-specific sources in the Discussion section.

Besides my general observations and suggestion, I have a few specific suggestions that should be addressed.

1.       Abstract: Please incorporate “increase in antioxidant’s enzymatic activity in response to metformin”.

2.       Introduction should be condensed (you can do so by avoiding repetition of metformin’s effects on restoration of aging-related pathologies or diseases)

3.       Line, 101; it would be better if the authors could mention the specific target residue in Nrf2 phosphorylated by AMPK.

4.        Lines, 109- 113 - too lengthy phrase- I suggest making it simple and readable.

5.       Please consider reducing the “Materials and Methods” section. This you can do by providing references for standard methods, such as cell culture, QPCR, ChIP assay, Nuclear and cytosolic isolation, Nrf2 transactivation assay etc.

6.       The Discussion section is rather long and could be condensed; I found there is repetition of reactivation or restoration of deteriorated antioxidant pathway by metformin, and author has emphasized this several times. I would like to suggest revising the “Discussion” section.

7.       The authors did not provide expression data on antioxidants, mRNA or protein in human LECs (however, they have provided enzymatic activities).

8.        There are some grammatical and typographic errors should be corrected throughout the manuscript.

9.       Finally, I would like suggest reducing the number of references.

Author Response

Cells- Manuscript Number:cells-1905259  (“Obligatory role of AMPK activation and antioxidant defense pathway in the regulatory effects of Metformin on cellular protection and prevention of  lens opacity” by Chhunchha et al.)

            Firstly, we would like to extend our thanks to the editor and   the reviewers for their careful evaluation of the manuscript, insightful comments, and constructive suggestions. In the revised manuscript, we believe that we have addressed the concerns and critiques raised by the reviewers.

            We trust that you will find that the issues raised by the reviewers have been adequately addressed and the overall quality of the manuscript has been substantially improved. We now believe it to be acceptable for publication in Cells. Detailed responses to the reviewers’ comments are provided below.

Responses to Reviewer’s Comments and suggestions

Reviewer 3 (Comments and Suggestions to the Authors):

            We thank the reviewer for providing suggestions, and these were helpful to improve the manuscript.

…………The potential effects of this work on the use of metformin for postponing or preventing age-associated disorders are very significant because they provide very clear evidence for the mechanism(s) by which metformin exerts its protective effects and for how it activates and controls the AMPK-Bmal1-Nrf2-mediated defense response. Therefore, I would like to recommend that the authors discuss this novel finding by referring lens-specific sources in the Discussion section.

            As suggested, we have now discussed the finding in the Discussion section. Please see page 35, Lines 1373-1380.

  1. Abstract: Please incorporate “increase in antioxidant’s enzymatic activity in response to metformin”.

            As suggested by the reviewer, we have now incorporated. Please see page 1, Lines, 17-18.

  1. 2.       Introduction should be condensed (you can do so by avoiding repetition of metformin’s effects on restoration of aging-related pathologies or diseases)

            In accordance with the reviewers’ comments, we have removed overlapping and/or repetitive sentences.

  1. 3.       Line, 101; it would be better if the authors could mention the specific target residue in Nrf2 phosphorylated by AMPK.

            We have now mentioned specific phosphorylation site of Nrf2 (Serine 550) in the text (Please see page 2 -3; lines, 98-102) and also placed the related reference no. 5, 36-41.

  1. 4.        Lines, 109- 113 - too lengthy phrase- I suggest making it simple and readable.

We have rephrased it to make it simple and short. Please see page 3; lines 107-110.

  1. 5.       Please consider reducing the “Materials and Methods” section. This you can do by providing references for standard methods, such as cell culture, QPCR, ChIP assay, Nuclear and cytosolic isolation, Nrf2 transactivation assay etc.

             We have shortened “Materials and Methods” as suggested by the reviewer.

  1. The Discussion section is rather long and could be condensed; I found there is repetition of reactivation or restoration of deteriorated antioxidant pathway by metformin, and author has emphasized this several times. I would like to suggest revising the “Discussion” section.

            We have reorganized the Discussion section by removing repetitive phrases and by adding discussion about our findings.

  1. The authors did not provide expression data on antioxidants, mRNA or protein in human LECs (however, they have provided enzymatic activities).

            We already have published fate or expression levels of these genes during oxidative stress and aging. We now placed the references to support our results. Please see Page 34; lines, 1297-1301.

  1. There are some grammatical and typographic errors should be corrected throughout the manuscript.

            Following the reviewer’s comments, we have revised the manuscript with special attention to correcting grammatical mistakes or typos. In addition, the manuscript has been reviewed by a professional editor.

  1. Finally, I would like to suggest reducing the number of references.

            In the revised manuscript, we have removed some references where required.
